# ATAC-Seq analysis reveals a widespread decrease of chromatin accessibility in age-related macular degeneration

Jie Wang[1], Cristina Zibetti [2], Peng Shang[1,8], Srinivasa R. Sripathi[1], Pingwu Zhang[1], Marisol Cano[1], Thanh Hoang[2], Shuli Xia[3,4], Hongkai Ji [5], Shannath L. Merbs[1], Donald J. Zack[1], James T. Handa[1], Debasish Sinha[1,8], Seth Blackshaw[1,2,3,6,7] & Jiang Qian[1]

Age-related macular degeneration (AMD) is a significant cause of vision loss in the elderly. The extent to which epigenetic changes regulate AMD progression is unclear. Here we globally profile chromatin accessibility using ATAC-Seq in the retina and retinal pigmented epithelium (RPE) from AMD and control patients. Global decreases in chromatin accessibility occur in the RPE with early AMD, and in the retina of advanced disease, suggesting that dysfunction in the RPE drives disease onset. Footprints of photoreceptor and RPE-specific transcription factors are enriched in differentially accessible regions (DARs). Genes associated with DARs show altered expression in AMD. Cigarette smoke treatment of RPE cells recapitulates chromatin accessibility changes seen in AMD, providing an epigenetic link between a known risk factor for AMD and AMD pathology. Finally, overexpression of *HDAC11* is partially responsible for the observed reduction in chromatin accessibility, suggesting that *HDAC11* may be a potential new therapeutic target for AMD.

[1] Department of Ophthalmology, Johns Hopkins University School of Medicine, Baltimore, MD 21205, USA. [2] Solomon H. Snyder Department of Neuroscience, Johns Hopkins University School of Medicine, Baltimore, MD 21205, USA. [3] Department of Neurology, Johns Hopkins University School of Medicine, Baltimore, MD 21205, USA. [4] Hugo W Moser Research Institute at Kennedy Krieger, Johns Hopkins University School of Medicine, Baltimore, MD 21205, USA. [5] Department of Biostatistics, Johns Hopkins Bloomberg School of Public Health, Baltimore, MD 21205, USA. [6] Center for Human Systems Biology, Johns Hopkins University School of Medicine, Baltimore, MD 21205, USA. [7] Institute for Cell Engineering, Johns Hopkins University School of Medicine, Baltimore, MD 21205, USA. [8] Present address: Departments of Ophthalmology, Cell Biology and Developmental Biology, University of Pittsburgh School of Medicine, Pittsburgh, PA 15224, USA. These authors contributed equally: Jie Wang, Cristina Zibetti. Correspondence and requests for materials should be addressed to S.B. (email: sblack@jhmi.edu) or to J.Q. (email: jiang.qian@jhmi.edu)

Age-related macular degeneration (AMD) is by far the most common cause of irreversible visual impairment in people over 60[1]. The estimated number of people with AMD in 2020 is 196 million, and will increase substantially with aging of the global population[2]. The disease is characterized by the early appearance of drusen, pigmentary abnormalities of the retinal pigment epithelium (RPE), and progressive photoreceptor dysfunction that is restricted primarily in the macula, a 6 mm diameter region of the fundus[3]. Although treatments aimed at inhibiting blood vessel growth can effectively slow the progression of the "wet" AMD, no useful treatments exist for the atrophic ("dry") form of the disease, which account for 90% of all AMD cases[4].

Currently, GWAS analysis has identified at least 34 AMD genetic risk loci involved in multiple pathways including regulation of the complement pathway and inflammation[5,6]. However, these genetic variants only explain a subset of AMD cases, suggesting a substantial role for environmental factors in the pathogenesis of AMD. Indeed, studies have linked variables such as cigarette smoking and obesity to AMD susceptibility, both of which are known to induce cellular stress and inflammation in a wide range of tissues[7,8]. Several groups have reported that DNA methylation changes in individual genes may be associated with AMD[9–12]. However, no comprehensive analysis of global chromatin accessibility changes associated with AMD progression has yet been reported. This in part, reflects the lack of widely-accepted animal models for AMD[4], as well as the difficulty in obtaining sufficient amounts of human pathological tissue for analysis. Here, we focus on less reported but more prevalent non-neovascular or "dry" AMD. We perform genome-wide chromatin accessibility studies and observe global and progressive decreases in chromatin accessibility associated with AMD onset and progression. Both cigarette smoke treatment and overexpression of the epigenetic regulator HDAC11 in human iPSC-derived RPE recapitulate the changes in chromatin accessibility. These findings suggest that global decreases in chromatin accessibility may play a critical role in the onset and progression of AMD.

## Results

**Landscape of chromatin accessibility in the retina and RPE.** In this study, we obtained 8 normal eyes from 5 donors, and 3 early dry, and 5 late dry, or geographic atrophic eyes from 5 AMD donors (Supplementary Table 1). We collected retina and pure RPE from the macular and peripheral regions of each donor eye, which altogether yielded a total of 19 normal, 9 early dry AMD, and 17 late geographic atrophic AMD-derived samples (Table 1). Cell-type specific gene expression analysis confirmed the high degree of purity of the retina and RPE samples that were used for

analysis (Supplementary Fig. 1a). Although the procurement time is slightly longer for normal samples, major characteristics including gender and age are comparable among normal and AMD samples. Disease severity was confirmed with visual examination by an expert observer (J.T.H.).

To study the global epigenetic landscape of AMD, we used the assay for transposase-accessible chromatin using sequencing (ATAC-Seq) to detect genomic chromatin accessibility, which depicts active (i.e., open) and inactive (i.e., condensed) chromatin[13]. We obtained an average of 78.5% mappability and 35.8 million qualified fragments per sample (Supplementary Table 2). ATAC-Seq data from two replicate samples, obtained from adjacent regions in peripheral retina of the same eye, showed high correlation ($R = 0.98$, Supplementary Fig. 1b), indicating that ATAC-Seq can reliably and reproducibly measure chromatin accessibility in these samples. In total, 78,795 high-confidence open chromatin regions (or peaks) were identified across all retinal samples, and 49,217 peaks were identified across all RPE samples, representing a total of 93,863 distinct peaks (Supplementary Data 1 and 2). Chromatin accessibility in the retina is overall higher than that of the RPE, potentially reflecting the much greater diversity of cell types in the retina relative to the RPE (Fig. 1a). Comparison of samples from the macular vs. peripheral retina, as well as the macular vs. peripheral RPE, showed broadly similar profiles of chromatin accessibility (Fig. 1a).

The data revealed categories of peaks that are either specific to, or shared between, the retina and RPE. For instance, a peak associated with RLBP1 is shared by the retina and RPE, whereas a peak associated with SLC1A2 is specific to the retina, and another peak within the SLC45A2 gene is RPE-specific (Fig. 1b). The peaks associated with typical housekeeping genes were often shared by the retina and RPE (Supplementary Fig. 1c). Furthermore, a small number of region-specific peaks (e.g., peaks in KCNC2 and ZIC1) are selectively detected in the macular and peripheral retina, respectively, while others (e.g., peaks in PKD1L2 and ALDH1A3) are selectively accessible in the macular and peripheral RPE (Fig. 1b). Overall, 39,394 (42.0%) ATAC-Seq peaks are shared by the retina and RPE, 38,625 (41.1%) peaks are retina-specific, and 15,844 (16.9%) peaks are RPE-specific (Fig. 1c). We identified 5,855 increased and 2,689 decreased peaks in the macular retina, relative to the paired retinal samples from the peripheral region (Supplementary Fig. 2a and Supplementary Data 3). Meanwhile, 432 increased and 959 decreased peaks were detected in the macular relative to peripheral RPE (Supplementary Fig. 2b and Supplementary Data 4). We observed that a great majority (81.7%) of peaks that are shared between the retina and RPE are also detected in other tissues (Fig. 1c). In contrast, only 4626 (12%) of retina-

| Table 1 The characteristics of ATAC-Seq samples | | | | | |
|---|---|---|---|---|---|
| **Variable** | **Tissue** | **Normal** | **Early AMD** | **Late AMD** | **P value\*** |
| No. of samples | Retina | 11 (58%) | 5 (56%) | 9 (53%) | 0.99 |
| | RPE | 8 (42%) | 4 (44%) | 8 (47%) | 0.99 |
| Region (macula) | Retina | 5 (45%) | 2 (40%) | 5 (56%) | 0.99 |
| | RPE | 3 (38%) | 2 (50%) | 6 (75%) | 0.38 |
| Gender (male) | Retina | 5 (45%) | 2 (40%) | 2 (22%) | 0.57 |
| | RPE | 5 (63%) | 1 (25%) | 2 (25%) | 0.41 |
| Age (years) | Retina | 84: 79 ~ 92 | 87: 82 ~ 87 | 90: 90 ~94 | 0.09 |
| | RPE | 88: 84 ~ 92 | 92: 85 ~ 97 | 90: 89 ~ 94 | 0.32 |
| Interval (hours)[a] | Retina | 10.1 | 5.3 | 6.6 | 0.01 |
| | RPE | 9.5 | 4.9 | 6.8 | 0.08 |

\*Fisher's exact test and one-way ANOVA were performed, respectively
[a]Interval indicates the time from death to procurement of eye

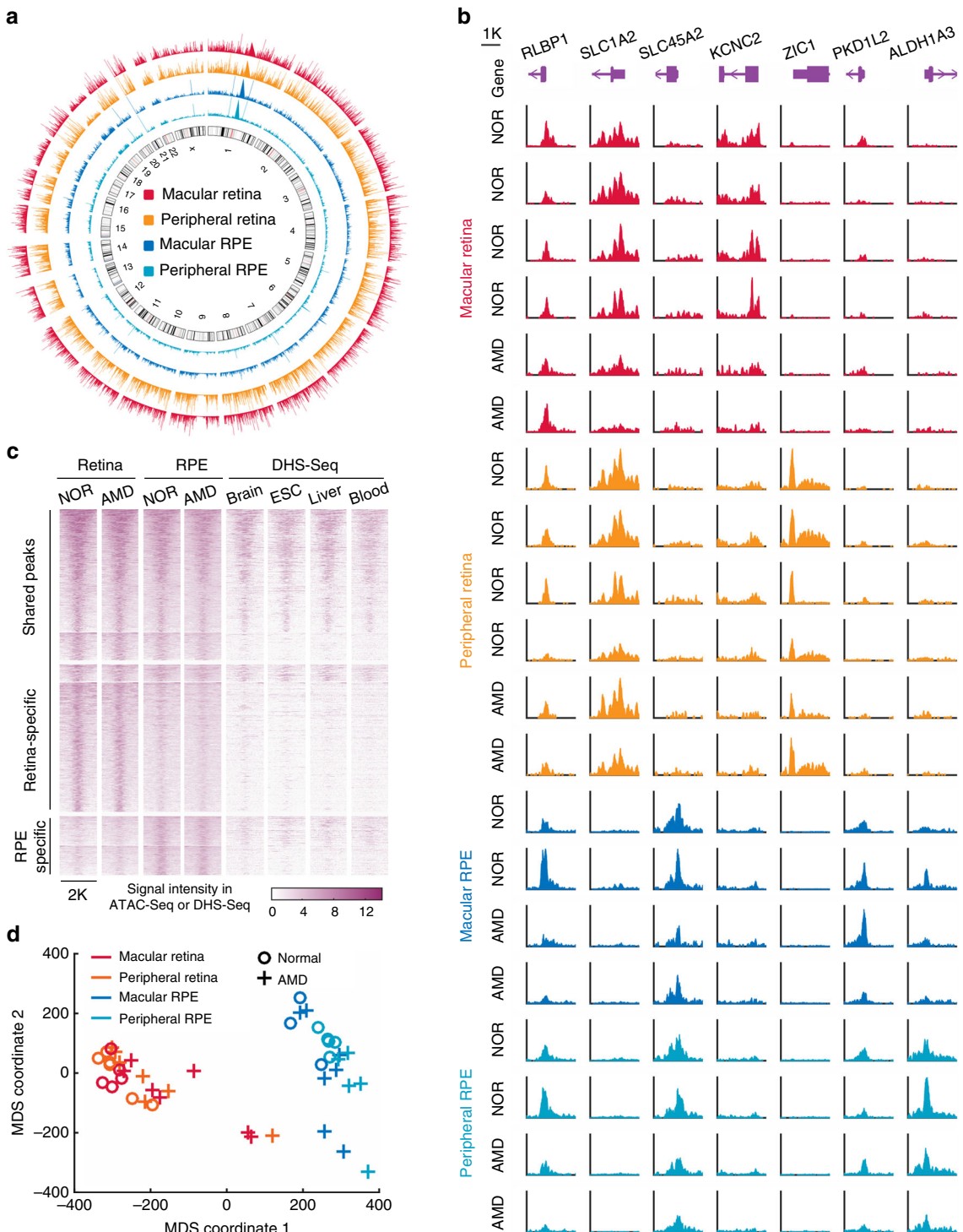

**Fig. 1** The landscape of chromatin accessibility in human retina and RPE. **a** Genome-wide chromatin accessibility of a control eye. **b** The instances of open chromatin in the retina and RPE from healthy controls (NOR) and AMD patients. **c** Specific and shared ATAC-Seq peaks in the retina and RPE. Each row represents one peak. The color represents the intensity of chromatin accessibility. Peaks are grouped based on K-means clustering and aligned at the center of regions. **d** Multidimensional scaling (MDS) of all retina and RPE samples

specific peaks and 7644 (48.2%) of RPE-specific peaks are detected in other tissues, implying that these peaks potentially represent highly tissue-specific *cis*-regulatory elements.

We then calculated the overall similarity of the ATAC-Seq profiles among all samples using multidimensional scaling. As expected, this analysis showed that the samples are clustered into two groups, one from the retina and the other from the RPE

(Fig. 1d). Moreover, most AMD samples are clearly separated from normal samples, especially for RPE, suggesting an extensive difference in chromatin accessibility between healthy and AMD tissues. An alternative and more detailed analysis showed that the separations between the AMD and controls are statistically significant (Supplementary Fig. 2c).

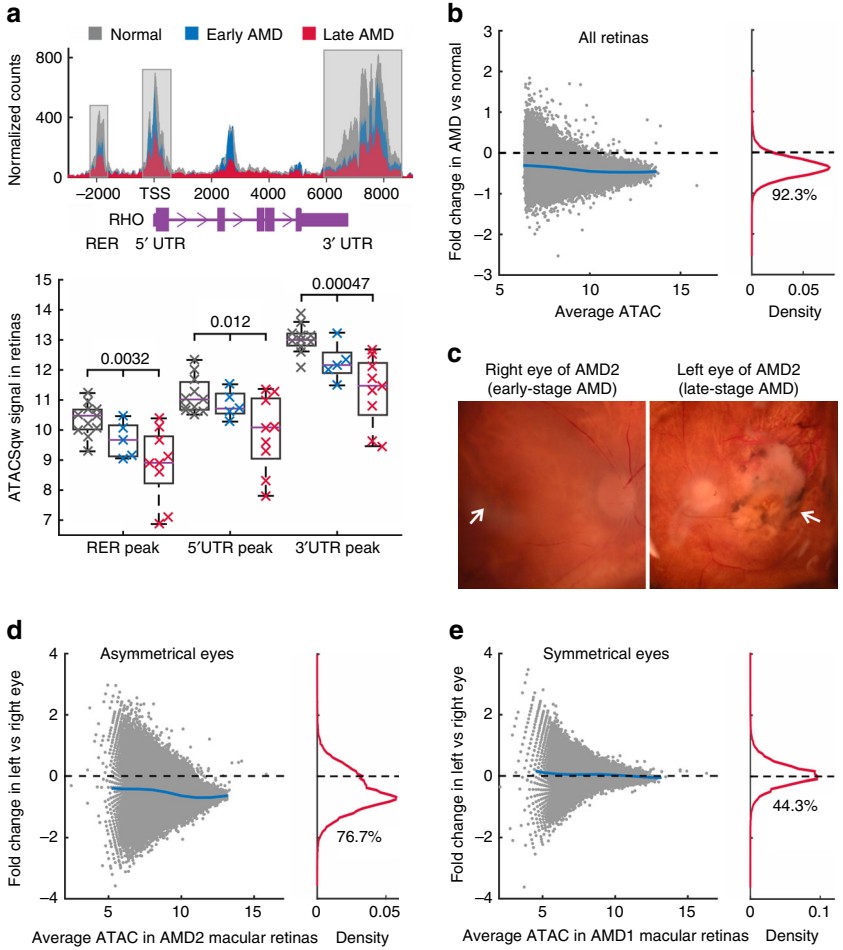

**Fig. 2** Changes of chromatin accessibility in AMD retinas. **a** The chromatin accessibility in regulatory regions of the rhodopsin gene *RHO*. Top panel shows average ATAC-Seq signal for each category. Bottom panel is the boxplot of log2-transformed ATAC-Seq signal for all samples (*n* = 11 for normal, 5 for early AMD, and 9 late AMD). One-way ANOVA test was performed. TSS, transcript start site. RER, rhodopsin enhancer region. UTR, untranslated region. **b** Changes of chromatin accessibility in AMD (*n* = 14) relative to normal (*n* = 11) in all retina samples. Each dot represents one ATAC-Seq peak. Blue line in the left panel indicates average fold changes of peaks with the same ATAC-Seq intensity. The percentage of reduced peaks is shown under the density curve in the right panel. **c** The microscopy of right and left eyes from one AMD patient with asymmetrical disease status. The left image shows early AMD (mild RPE pigmentary changes) while the right image shows geographic atrophy. Arrows indicate the macular regions. Note that human macula has a diameter of around 6 mm. **d** ATAC-Seq signal changes in right and left macular retinas from the AMD patient with asymmetrical disease status. **e** Accessibility changes in one AMD patient whose eyes are at the same (symmetrical) disease stage

**Chromatin accessibility is broadly decreased in AMD samples.** To explore the impact on AMD, we analyzed the differences in chromatin accessibility between normal and AMD retinas. When comparing the accessibility profiles, we noticed substantial quantitative differences in peak signal between normal and AMD retina samples. For example, in three known regulatory regions of the rhodopsin gene *RHO*, chromatin accessibility is progressively decreased from normal to early-stage, and then to late-stage AMD (*P* < 0.05, Fig. 2a). By comparing the signal for each peak in healthy and AMD samples from both macular and peripheral retinas, we observed that 72,689 (92.3%) peaks have reduced chromatin accessibility in AMD (Fig. 2b). These quantitative differences in chromatin accessibility do not result from the process of normalizing ATAC-Seq data because different normalization approaches gave similar results (Supplementary Fig. 2d). Moreover, we separated retina samples into two groups from the macular and peripheral regions. Relative to the peripheral region, we observed a more intense global decrease in chromatin accessibility from the macular (94.5%) than peripheral (79.9%) region of AMD retina (Supplementary Fig. 2e and 2f).

To extend this observation, we obtained a pair of eyes from a donor whose AMD status was asymmetrical, with the right eye showing early-stage AMD, and the left eye showing late-stage dry AMD (Fig. 2c). By comparing these eyes, we excluded the contribution of potential genetic and environmental differences that might complicate the analysis of epigenetic changes associated with AMD progression. Interestingly, a large number (76.7%) of peaks in the macular retina from the more severely affected eye had decreased intensities relative to the less severely affected eye (Fig. 2d). We then compared additional 5 pairs of eyes as "controls". For these donors whose left and right eyes were at the same disease stage, the chromatin accessibility profiles were highly symmetrical in their macular retinas (Fig. 2e and Supplementary Fig. 3a). This analysis confirmed that a widespread decrease in chromatin accessibility is associated with AMD progression.

Next, we analyzed changes of RPE chromatin accessibility in AMD. In all RPE samples, a great number (91.6%) of peaks showed the reduced intensity in AMD relative to normal samples (Fig. 3a). This reduction in intensity associated with AMD RPE was observed in both macular and peripheral regions (88.6% for

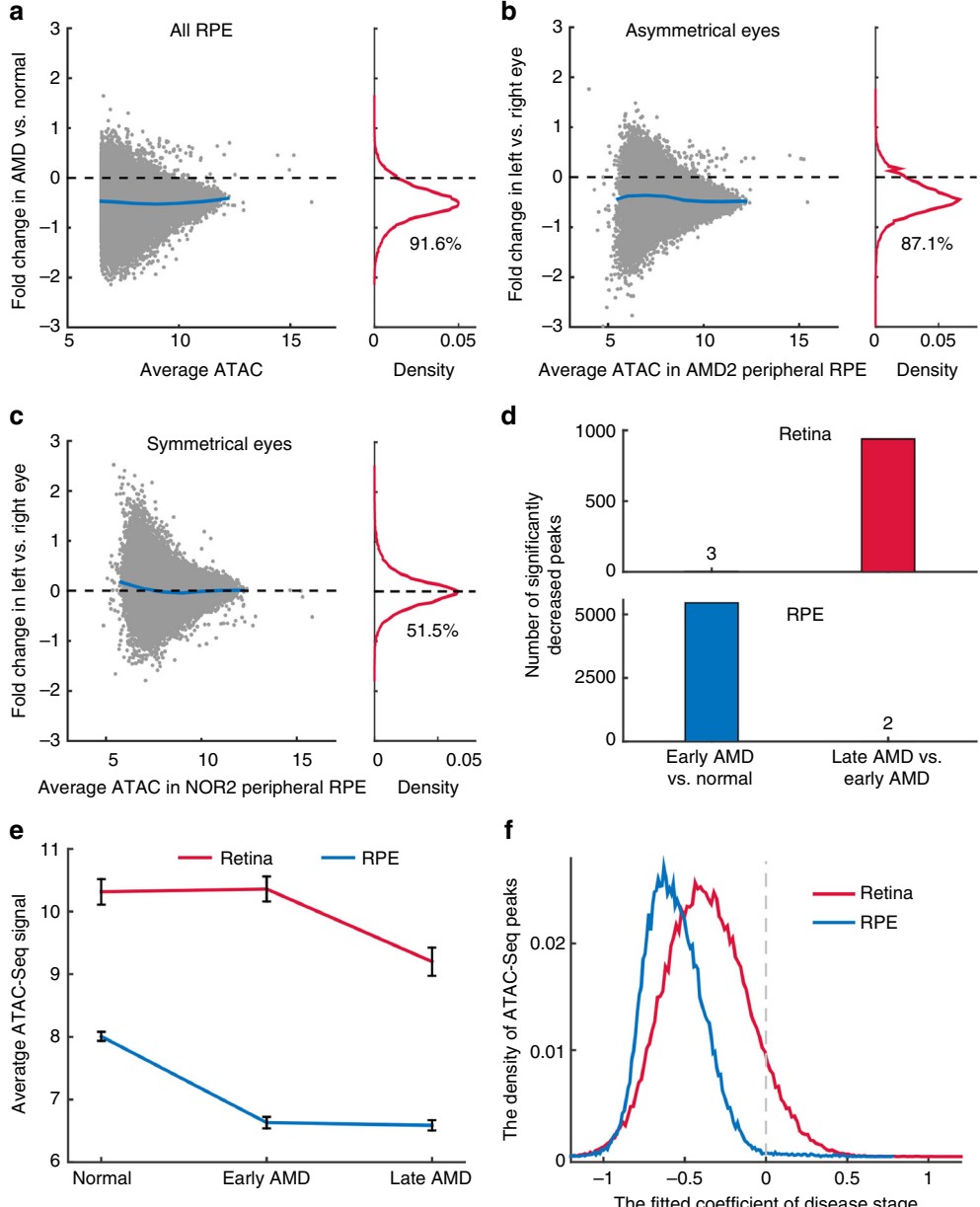

**Fig. 3** Changes in chromatin accessibility in the RPE and at different disease stages. **a** Changes of chromatin accessibility in AMD ($n = 12$) relative to normal ($n = 8$) for all RPE samples. Blue line in the left panel indicates average fold changes of peaks. The percentage of reduced peaks is showed under the density curve. **b**, **c** Changes of chromatin accessibility in the RPE from donors whose eyes are at the different (asymmetrical) or the same (symmetrical) stage of disease. **d** The number of peaks in the retina and RPE with significantly decreased accessibility for early AMD ($n = 5$ for the retina, 4 for the RPE) vs. normal ($n = 11$ for the retina, 8 for the RPE) or late AMD ($n = 9$ for the retina, 8 for the RPE) vs. early AMD (late stage). **e** Average signal of ATAC-Seq peaks with differential accessibility at any stage of AMD. Error bars represent the standard error of mean. **f** The density curves of the stage coefficients in the fitting model of retina and RPE ATAC-Seq peaks

macula and 94.1% for periphery showed reduced ATAC-Seq signal) (Supplementary Fig. 3b and 3c). In the RPE from the patient who showed different stages of AMD between eyes, the intensities of 42,860 (87.1%) peaks were reduced in the more severely affected left eye (Fig. 3b). In contrast, a symmetrical distribution was observed in donors where both eyes were at the same disease stage (Fig. 3c and Supplementary Fig. 3d). If only one sample from each of ten donors was included, a similar global decrease in chromatin accessibility was observed in the retina and RPE (Supplementary Fig. 3e and 3f). Large genomic domains (on the order of 1-2 Mb) were also found to have globally differential chromatin accessibility (Supplementary Fig. 4). Taken together,

our data show a widespread decrease in chromatin accessibility that is observed in both the retina and RPE from AMD patients.

**Decreased chromatin accessibility at different stages of AMD.**
We further set out to identify changes in ATAC-Seq peak intensity that were associated with disease stage in both the retina and RPE. When we compared 5 retinal samples obtained from early-stage AMD to 11 retinal samples obtained from healthy controls, we observed only 3 statistically significant decreases in peak intensity (Fig. 3d and Supplementary Fig. 5a). However, by comparing 9 late-stage AMD retinal samples to these same 5 early-stage retinal samples, we observed 939 peaks with

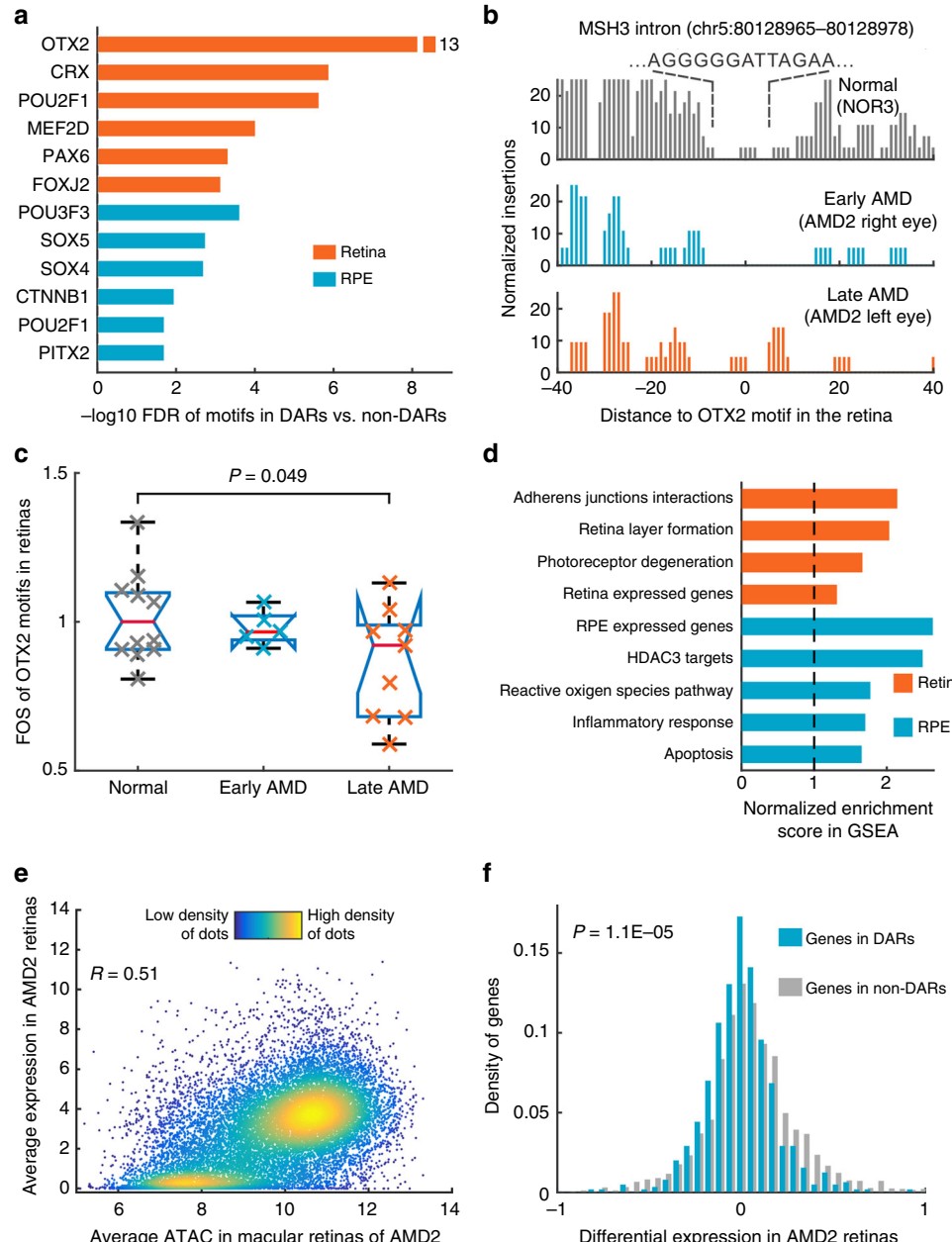

**Fig. 4** The regulation and expression associated with DARs in the retina and RPE. **a** Enriched TF motifs in footprints within DARs. **b** An example of footprint of *OTX2* in *MSH3* gene. An *OTX2* footprint located at the intron of *MSH3* was observed in the normal sample, decreased in the early-stage AMD, and absent from late-stage AMD sample. **c** Footprint occupancy scores (FOS) for *OTX2* motifs in normal, early-stage, and late-stage AMD retinas. Student's *t*-test was performed between normal samples and late-stage AMD samples. **d** Significantly enriched functions of DAR nearby genes from gene set enrichment analysis (GSEA). **e** The relationship of chromatin accessibility and RNA-Seq measured gene expression in retinas. **f** The density of differential expression in left vs. right retinas of the AMD patient. P value for Student's *t*-test is shown

significantly decreased intensity, suggesting that the chromatin accessibility changes in the retina occur primarily during late stages of disease.

In contrast, when we compared 4 RPE samples from early-stage AMD to RPE samples from 8 healthy controls, we observed 5458 significantly decreased peaks, but observed only 2 significantly decreased peaks when these same early-stage samples were compared to 8 RPE samples from late-stage AMD (Fig. 3d and Supplementary Fig. 5a). Likewise, when averaging the intensities of significantly decreased peaks at any stage of AMD, we found a striking decrease of chromatin accessibility in the RPE at an earlier disease stage than that observed in the retina (Fig. 3e). This observation fits with the widely accepted theory that changes in

RPE function trigger AMD[14], and suggests that epigenetic changes in RPE cells might be a critical factor that regulates disease onset.

**AMD-associated changes in gene regulatory networks**. We next sought to determine the functional consequence of the differentially accessible regions (DARs) that were observed in normal and AMD samples. To define statistically significant DARs, we used a linear regression model to take into account the potential effects from other confounding factors such as topographical differences (macula vs. periphery), age, gender, and procurement interval. The model estimated the relative contributions from these factors

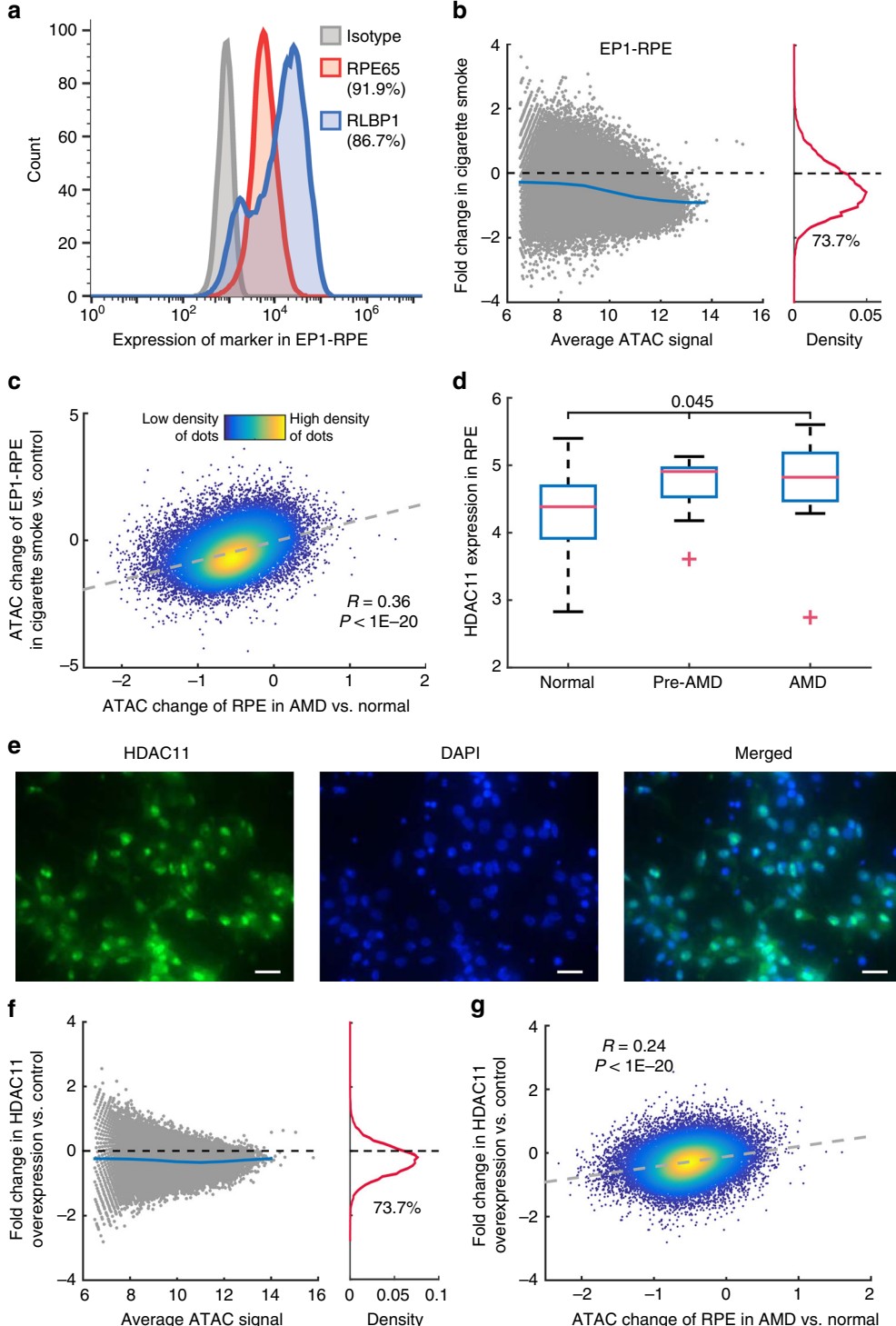

**Fig. 5** Chromatin accessibility changes in cigarette smoke-treated or *HDAC11*-overexpressed RPE cells. **a** Flow cytometric analysis of the expression of RPE specific markers RPE65 and RLBP1 from 12-week-old iPSC-derived RPE monolayers. Isotype was used as the control for gating strategy. **b** Changes in chromatin accessibility after cigarette smoke treatment. The average intensities of two replicates were used in the analysis. The percentage of reduced peaks is shown under the density curve in the right panel. **c** Comparison of chromatin accessibility changes in AMD RPE and cigarette smoke-treated RPE cells. **d** *HDAC11* expression in peripheral RPE at different disease stages ($n = 46$ for normal, 9 for pre-AMD, and 14 for AMD). One-way ANOVA test was performed. The red symbol ' + ' indicates the outlier. The data are from NCBI GEO GSE29801. **e** *HDAC11* and DAPI staining in the RPE cell. Scale bar, 50 μm. **f** Changes in chromatin accessibility with *HDAC11* overexpression. **g** Comparison of accessibility changes in AMD RPE and *HDAC11*-overexpressed RPE cells. The gray dashed line is the fitting line. R is Pearson's correlation coefficient

and the effects of disease stage (normal, early, and late AMD) to variations in peak intensity. Our analysis suggested that the peaks in macular retinas are more likely to be reduced than those in peripheral retinas (Supplementary Fig. 5b). Notably, a longer procurement interval leads to smaller peaks. Given that the procurement interval of AMD samples is slightly shorter than normal samples (Table 1), it is highly unlikely that an altered procurement interval leads to the decreased peak intensity in AMD samples that is observed in this study. Most importantly, the coefficients for disease stage are significantly negative for a large number of the peaks in the retina (38,520 peaks, 48.9%, FDR < 0.05) and in the RPE (41,168 peaks, 83.7%, Fig. 3f and Supplementary Fig. 5b and 5c), suggesting that late stages of disease are associated with lower peak intensity.

For the retina and RPE, we chose the top 5000 peaks with significantly negative coefficients of disease stage as DARs (set FDR < 0.01 and ranked by the coefficients, Supplementary Data 5 and 6). We examined the genomic location of these DARs and found that retinal DARs are enriched in intergenic regions (Supplementary Fig. 6a). RPE DARs, in contrast, are enriched in promoters. By checking whether transcription factor (TF) binding was affected in the retina and/or RPE, we observed 22 and 13 TF motifs that are strongly enriched in the retinal and RPE DARs, respectively (Fig. 4a and Supplementary Table 3). For example, the binding motifs of OTX2 and CRX, factors known to play an important role in controlling gene expression in photorecep-tors[15,16], are enriched in AMD retinas. Moreover, OTX2 showed a significantly decreased footprint in DARs for comparison of late-stage AMD to normal samples (Fig. 4b, c). This pattern confirmed that chromatin accessibility of OTX2 target sites is decreased in retinal samples with AMD disease, suggesting that reduced target sites binding by retina and RPE-specific TFs play a critical role in AMD pathogenesis.

**Genes in DARs show altered expression in AMD**. We further tested whether the expression levels of genes associated with DARs are more likely to be altered in AMD. We first checked that DAR-associated genes in both the retina and RPE were highly enriched for genes that were selectively expressed in each tissue (Fig. 4d). Moreover, DAR-associated genes in the retina were substantially more likely to regulate retinal layer lamination and photoreceptor survival, while DAR-associated genes in the RPE were more likely to regulate the inflammatory response and apoptosis, which are important biological processes in AMD (Fig. 4d). We also observed that housekeeping genes are depleted from DAR-associated genes in the retina and overrepresented in DAR-associated genes in the RPE (Supplementary Fig. 6b). These housekeeping genes associated with DARs in the RPE are involved in mitochondrion and cellular response to stress.

Using RNA-Seq data obtained from the patient with differential AMD stages between eyes, we observed that ATAC-Seq peak intensity was highly correlated with gene expression in both the retina and RPE (Fig. 4e and Supplementary Fig. 6c). In the patient with asymmetric AMD progression, DAR-associated genes were significantly more likely to be downregulated in late-stage relative to early-stage AMD ($P = 1.1 \times 10^{-5}$, Fig. 4f and Supplementary Fig. 6d). These results suggest that altered chromatin accessibility in binding sites of retina and RPE-enriched TFs leads to reduced expression of associated genes in AMD.

**Relationship to AMD-associated genetic variants**. To examine whether the observed changes in chromatin accessibility resulted from AMD-associated genetic variants, we compared the distribution of DARs to that of genetic variants linked to AMD

susceptibility by GWAS analysis[5]. For each DAR, we tested whether it overlapped with one or more AMD-associated SNPs identified by GWAS. Interestingly, we observed that very few of AMD-associated SNPs were covered by DARs in genomic location. There are <0.1% of all DARs in the retina, and <0.2% of DARs in the RPE, overlapped with AMD-associated SNPs (Supplementary Fig. 6e). Even if we extend a 5 kb window in each direction of DARs, the proportion of DARs overlapped with AMD-associated SNPs was still low (<0.3% for the retina and <0.4% for the RPE). For comparison, we also examined the fractions of non-DARs and non-peaks that overlapped with AMD-associated SNPs, and found that the overlap was comparable with those of DAR regions in both retina and RPE. These data imply that the observed differences in chromatin accessibility are unlikely due to local AMD-associated genetic variants.

**Chromatin accessibility changes induced by cigarette smoke**. Since cigarette smoking is the strongest environmental risk factor for AMD[17], we tested whether cigarette smoke treatment of cultured human RPE cells could trigger similar changes in chromatin accessibility from AMD samples. Terminally differentiated iPSC-derived RPE cells were examined by flow cytometric analysis of RPE-specific markers (Fig. 5a). In addition, we performed RNA-Seq in the cultured RPE cells and compared its profile with published expression profiles in normal human RPE cells[18]. We confirmed that the iPSC-derived RPE cells showed a great similarity in gene expression profile to normal RPE cells ($R = 0.81$, Supplementary Fig. 7a). Furthermore, ATAC-Seq was performed on the cultured RPE cells and showed a broadly similar pattern of peak distribution to an average ATAC-Seq profile of all healthy samples from RPE tissue ($R = 0.83$, Supplementary Fig. 7b). Finally, we analyzed the karyotype of iPSC-derived RPE cells and confirmed that they are normal cells (Supplementary Fig. 7c). In summary, these analyses demonstrate the fidelity of the iPSC-derived RPE cells relative to normal RPE cells.

The iPSC-derived RPE cells were exposed to cigarette smoke extract and ATAC-Seq profiling was performed before and after the smoke treatment. A global decrease in chromatin accessibility was observed in RPE cells after treatment (Fig. 5b). Comparison of two biological replicates in cigarette smoke-treated iPSC-derived RPE cells showed that the effect is reproducible (Supplementary Fig. 7d). More importantly, when comparing cigarette smoke treated iPSC-derived RPE cells with RPE tissue from AMD patients, we found that the changes in chromatin accessibility are highly correlated (Fig. 5c; $R = 0.36$, $P < 10^{-20}$). The observation was confirmed in one additional iPSC-RPE cell line (Supplementary Fig. 7e-7h). These results indicate that cigarette smoke treatment in RPE cells induces a widespread decrease of chromatin accessibility that is much like that seen in AMD.

*HDAC11* overexpression causes reduced chromatin accessibility. Next, we attempted to identify the genes that could induce similar changes in chromatin accessibility observed during AMD progression. To do so, we analyzed a previously published collection of microarray data from AMD samples[19], along with our own RNA-Seq analysis, to identify differentially expressed genes that are known to regulate chromatin accessibility. Among all histone deacetylase (HDAC) genes, we found that three of them (*HDAC10, HDAC11, SIRT1*) were significantly differentially expressed between AMD and controls (Supplementary Table 4). *HDAC11*, however, both showed significantly increased expression in the RPE during early disease stages (Fig. 5d), and is also predominantly localized to the nucleus of RPE cells (Fig. 5e).

Moreover, cigarette smoke treatment of iPSC-derived RPE cells also increased *HDAC11* expression by western-blot analysis (Supplementary Fig. 8a). Increased *HDAC11* expression was consistent with a global reduction in H3K27ac levels (Supplementary Fig. 8b).

We then tested whether overexpression of *HDAC11* induces a broad decrease in chromatin accessibility. First, we confirmed that *HDAC11* was overexpressed in plasmid transfected RPE cells (Supplementary Fig. 8c). We then examined the chromatin accessibility profile in the monolayers of *HDAC11*-overexpressing RPE cells, and observed a widespread decrease of chromatin accessibility in *HDAC11*-overexpressing cells relative to cells transfected with empty vector (Fig. 5f). Interestingly, the changes induced by *HDAC11* overexpression were strongly consistent with changes seen in the RPE of AMD patients (Fig. 5g). Moreover, H3K27ac was reduced following *HDAC11* overexpression in RPE cells (Supplementary Fig. 8d). These results suggest that *HDAC11* overexpression may be partially responsible for the global decreases of chromatin accessibility associated with AMD progression.

## Discussion

It is known that environmental factors contribute to the development of AMD[20,21]. These environmental factors may alter epigenetic marks, which in turn can lead to broad biological consequence[22]. DNA methylation in blood or retina has been studied in AMD[9–12], with one example of an AMD-associated change being the hypomethylation of the *IL17RC* promoter that is observed in peripheral blood leukocytes[10]. However, the finding remains controversial[11]. Overall, the changes of DNA methylation in AMD are quite subtle.

In this work, we profiled changes in chromatin accessibility that occur during AMD progression in RPE and retina. While most studies have focused on neovascular AMD, our study focused on changes at the highly prevalent early stage and the less reported late, atrophic stage of AMD. We report a comprehensive analysis of changes in chromatin accessibility in AMD. These changes in chromatin accessibility are seen first in the RPE, and then later in the retina. Likewise, we observed greater changes in the macular retina than in the peripheral retina. These data fit with the typical pattern of AMD progression, where changes in the RPE precede the dysfunction and death of macular photoreceptors[14].

Several lines of evidence suggested that the changes in chromatin accessibility are not simply due to cell death in the retina or RPE. First, a diverse proportion of cell type-specific genes (e.g., rods, cones, and Müller glia) were associated with decreased ATAC-Seq peaks (Supplementary Fig. 8e), suggesting that the decrease of chromatin accessibility is not simply due to photoreceptor death. Second, the genes associated with DARs are enriched for specific TFs and cellular functions, suggesting that the regions for reduced chromatin accessibility are selective. Third, both cigarette smoke treatment and *HDAC11* overexpression mimic the effect of AMD on chromatin accessibility. Together, these suggest that cell death in AMD is unlikely to cause the observed global reduction in chromatin accessibility.

A relative small number of samples were used for our analysis. However, since the changes in chromatin accessibility are both large and widespread, we are able to draw a meaningful conclusion. Indeed, multiple analysis approaches (e.g., global comparison, left and right eye comparison, different disease stage comparison, and the linear regression model) consistently confirmed the scientific rigor of this observation. Our sample size was comparable to those used in other epigenomic studies[10,23,24],

suggesting that some epigenetic changes might be dramatic and require relatively small number of samples.

Our study demonstrated that upregulated *HDAC11* expression might be partially responsible for the observed changes in chromatin accessibility in AMD. However, the effect of *HDAC11* on chromatin accessibility is limited, suggesting that other factors (e.g., HATs) may also contribute to the observed DARs. Beyond changes in expression of these general chromatin and DNA modification enzymes, which do not possess sequence specificity, we hypothesize that altered expression of specific transcription factors (TFs) that play a role in guiding these enzymes to specific genomic loci may also account for the observed changes in chromatin accessibility in AMD.

We hypothesize that global changes in chromatin accessibility may be seen in many other diseases. Indeed, large-scale changes in chromatin accessibility have been previously reported in metastatic cancer, where globally increased chromatin accessibility appears to directly drive disease progression[25]. Cancer cells show generally high levels of metabolic activity, and these changes may partially reflect this fact[26]. In contrast, neurodegenerative diseases such as AMD are associated with decreased metabolic activity from early stages onward[27]. The resulting reduction in cellular levels of acetyl-CoA, an essential cofactor for histone acetylation, that occurs during disease progression may be a common mechanism that contributes to the observed changes in chromatin accessibility[28]. Our data raise the possibility that global, quantitative reduction in chromatin accessibility may also be observed in other retinal dystrophies, and for neurodegenerative diseases in general.

AMD risk loci are not significantly over-represented in the identified DARs, suggesting that the observed differences in chromatin accessibility are unlikely to result from local AMD-associated genetic variants. However, our study does not exclude the possibility that the chromatin accessibility is associated with other genetic variants. To fully investigate the possible interplay between the two factors, simultaneous measurement of chromatin accessibility and genetic variants will need to be conducted in a much larger number of samples.

ATAC-Seq is a widely used approach to detect chromatin accessibility. However, one potential confounding variable is nuclear envelope permeability, which could influence observed ATAC-Seq signals. Since this analysis was performed using nuclear extracts, it is conceivable that AMD might induce differences in nuclear permeability. While multiple lines of evidence presented in this study strongly support our interpretation that ATAC-Seq signals reflect chromatin accessibility, it remains possible that AMD might affect nuclear envelope permeability.

While decreased chromatin accessibility may be a hallmark of AMD, at present we cannot determine whether it represents a harmful change that compromises cell viability, or represents an adaptive response that may allow diseased cells to continue to function even in the face of reduced metabolism. A toxic role of decreased chromatin accessibility would fit with extensive data suggesting that pharmacological inhibition of histone deacetylases, which also leads to a global decrease in chromatin accessibility[29,30], is neuroprotective[31,32]. Indeed, broad-spectrum *HDAC* inhibitors that also target *HDAC11* have been shown to protect against ischemic damage in the retina[33–35], and in certain rodent models of photoreceptor degeneration[36,37]. However, in other cases, *HDAC* inhibition is neurotoxic[38], suggesting that these effects may depend on the specific HDACs that are expressed, their level of expression, or the stage of disease. Identifying the precise molecular mechanism that mediates the global changes in chromatin conformation that we observed in AMD will provide key information to answer this question, and

may point the way towards new therapeutic targets for preventing or slowing AMD progression.

## Methods

**Human samples**. Fresh postmortem eyes were processed within 14 h after death when obtained from Eye Banks (Portland, USA) and National Disease Research Interchange (Philadelphia, USA). The study was approved by Johns Hopkins Institutional Review Boards. All donors gave their written informed consent. The medical records do not contain any individually identifiable information. Donor information is summarized in Supplementary Table 1. The disease conditions were determined by medical record, and the eye globes were further examined by an experienced retinal physician with expertize in AMD (J.T.H.). The retinas were defined as normal when there were no abnormalities observed using a dissecting microscope. Early-stage AMD was defined by the presence of any RPE pigmentary changes and/or large-size drusen (>125 μm diameter). Late-stage AMD was defined by areas of geographic atrophy due to loss of the RPE. In this study, we only included dry AMD and excluded wet AMD. For each eye, we separated the retina and RPE and then obtained a punch (6 mm diameter) of retina or RPE tissue in each of macular and peripheral regions. Under direct visualization with a dissecting microscope, the RPE was mechanically separated from the choroid. We obtained paired eyes from some donors, and just one eye from other donors (see Table 1 and Supplementary Table 1).

**ATAC and RNA sequencing**. Biopsy punches of fresh retina and RPE tissues were re-suspended in cold PBS according to ATAC-Seq protocol[39]. Chromatin was extracted and processed for Tn5-mediated tagmentation and adapter incorporation, according to the manufacturer's protocol (Nextera DNA sample preparation kit, Illumina®) at 37 °C for 30 min. Reduced-cycle amplification was carried out in the presence of compatible indexed sequencing adapters. The quality of the libraries was assessed by a DNA-based fluorometric assay (Thermo Fisher Scientific[TM]) and automated capillary electrophoresis (Agilent Technologies, Inc.). Up to 3 samples per lane were pooled and run on a HiSeq2500 Illumina sequencer with a paired-end read of 50 bp.

Total cellular RNA was purified using the Qiagen RNAeasy Mini kit and samples with RNA integrity number ≥7 were further processed for sequencing. Libraries were prepared using Illumina TruSeq RNA Sample kit (Illumina, San Diego, CA) following the manufacturer's recommended procedure. Briefly, total RNA was denatured at 65 °C for 5 min, cooled on ice, purified and incubated at 80 °C for 2 min. The eluted mRNA was fragmented at 94 °C for 8 min and converted to double stranded cDNA, end repaired, A-tailed, and ligated with indexed adapters and run on a MiSeq Illumina sequencer. The quality of the libraries was also assessed by RNA-based fluorometric assay (Thermo Fisher Scientific[TM]) and automated capillary electrophoresis (Agilent Technologies, Inc.).

**Mapping and normalization of ATAC-Seq**. After removing adaptors using Trimmomatic[40], 50 bp paired-end ATAC-Seq reads were aligned to the human reference genome (GRCh37/hg19) using Bowtie2 with default parameters[41]. After filtering reads from mitochondrial DNA and the Y chromosome, we included properly paired reads with high mapping quality (MAPQ score >10, qualified reads) through SAMTools for further analysis[42]. Duplicate reads were removed using the Picard tools MarkDuplicates program (http://broadinstitute.github.io/picard/).

ATAC-Seq peak regions of each sample were called using MACS2 with parameters --nomodel --shift -100 --extsize 200[43]. Blacklisted regions were excluded from called peaks (https://www.encodeproject.org/annotations/ENCSR636HFF/). To generate a consensus set of unique peaks, we next merged ATAC-Seq peaks for which the distance between proximal ends was less than 10 base pairs. In total, we identified 308,019 peaks from retina samples and 208,592 from RPE samples. For each retina and RPE sample, the fragments were counted across each peak region using HTSeq[44]. We further calculated the normalized fragments ($C_N$) by dividing the raw fragments ($C_R$) by the library size ($S_L$) using the formula: $C_N = \log_2\left(\frac{C_R}{S_L} \times 100,000,000 + 1\right)$. In the study, we used the count of qualified fragments as the total library size for each sample. For downstream analysis, we only included those peak regions with average normalized signal ≥6.5 (around the value of 75% quantile) across retina or RPE samples. We obtained 78,795 peaks for retina samples and 49,217 peaks for RPE, comprising 93,863 distinct peaks in total. To validate whether different normalization approaches have altered these results, we also used the count of the properly paired fragments as the library size for normalizing fragment counts.

**K-means clustering and multidimensional scaling**. The circle plot was performed to visualize genomic ATAC-Seq peaks using Circos[45]. To compare chromatin accessibility of retina and RPE with other tissues, we mapped chromatin accessibility of 125 cells or tissues measured by DHS-Seq to ATAC-Seq peak regions[46]. DHS-Seq data were downloaded from ENCODE project (https://genome.ucsc.edu/ENCODE/index.html). K-means clustering was used to divide ATAC-Seq peaks into tissue-specific and shared groups. To present a two-dimensional distribution of the retina and RPE samples, we performed multi-dimensional scaling (MDS), in which all pairwise Euclidean distances were

calculated as the distance metric. The distances in MDS represent the similarity of samples. This analysis was carried out using the R software package.

**Analysis of differential chromatin accessibility**. An MA plot (log₂ fold change vs. mean average) was used to visualize changes in chromatin accessibility for all peaks. For ATAC-Seq peaks, we accessed the significant change of chromatin accessibility between different groups using edgeR[47]. The total count of the qualified fragments in each sample was used as the library size. It was defined as significantly changed if the peak showed |log₂ fold change| > 0.8 and FDR < 0.05. We compared samples in the macular region from healthy tissues to paired samples from the peripheral region to identify region-specific ATAC-Seq peaks. For the comparison among three groups of normal, early AMD, and late AMD samples, an ANOVA-like test was performed to identify peaks with significant differences (FDR < 0.01 was considered significant).

In order to estimate the relative contribution of disease stage (normal, early-stage, and late-stage) to the change of ATAC-Seq signal, compared to other potentially confounding factors, we conducted linear regression modeling in which the normalized ATAC-Seq fragments were taken as the dependent variable. The confounding factors include the region sampled (macula and periphery), gender (male and female), age, and procurement interval. Data fitting was performed separately for retina and RPE samples.

The formula for linear regression is following: ATAC-Seq signal = $\alpha_0 + \alpha_1 \times$ stage (0, 1, and 2 separately for normal, early-stage and late-stage) + $\alpha_2 \times$ region (−1 and 1 separately for periphery and macula) + $\alpha_3 \times$ gender (−1 and 1 separately for female and male) + $\alpha_4 \times$ age (years) + $\alpha_5 \times$ interval (hours) + ε. In the formula, the parameter $\alpha$ is the fitted coefficient for each variable, which represents the effect of variable on the change of chromatin accessibility. The parameter ε represents random noise. This was performed using the function of linear models 'lm' in the R platform.

From the peaks that showed significant association between change in accessibility and disease stage (using cutoff of FDR < 0.01), we selected the top 5000 peaks that showed the strongest negative correlation with disease stage as differentially accessible regions (DARs) separately for retina and RPE. The remaining ATAC-Seq peaks were classified as non-DARs.

**Genomic features and function enrichment of DARs**. We used ANNOVAR to associate ATAC-Seq peaks with the nearest genes[48]. ATAC-Seq peaks were assigned to four categories: promoter-proximal, exonic, intronic, 3′UTR and intergenic. Promoter is defined as the region within 2 kb of the reference transcript start site (TSS), as determined by the from UCSC genome browser (https://genome.ucsc.edu/). Peaks located in 5′UTR were also taken as promoter-proximal peaks. Peaks located within 2 kb of transcript end site were included with the 3′UTR peaks. Gene-proximal peaks include peaks within 2 kb of the gene body.

Using a linear model, we obtained the coefficient of disease stage for each ATAC-Seq peak and its associated gene. We then ranked genes based on the corresponding coefficient of disease stage. If multiple ATAC-Seq peaks were mapped to the same gene, we assigned the lowest coefficient to the gene. Function enrichment of accessibility change-ranked genes were further performed using GSEA software[49]. Besides GSEA database, sets of genes differentiatially regulating during the course of photoreceptor degeneration, as well as retina-expressed genes and RPE-expressed genes, were obtained from previous studies[19,50,51]. Housekeeping genes used in the study are from a previous report[52]. A similar analysis of GSEA was performed to identify hot spots which are enriched for DAR-associated genes.

**Motifs and footprints of DAR-related transcription factors**. To identify DAR-related transcription factors (TFs), we obtained 1043 position weight matrices of TF motifs from the TRANSFAC database[53]. To limit our analysis to TFs that were expressed in RPE or retina, we filtered out low expressed TFs in both normal and AMD tissues based on microarray data from GSE29801[19]. We thus identified 485 TF motifs in the retina dataset and 521 motifs in the RPE dataset. We first scanned all potential TF binding motifs (P value < 1.0⁻⁴) across the human genome using FIMO[54]. We used DNase2TF to identify all potential functional TF binding site using footprinting analysis in open regions (ATAC-Seq peaks) in both the retina and RPE samples[55]. We next identified TF footprints associated with both DARs and non-DARs. A hypergeometric test was further performed to assess the enrichment of TF footprints in DARs. Similar enrichment analyzes were conducted for gene-proximal and distal (or intergenic) DARs, using gene-proximal and distal non-DARs as control.

To compare TF footprints from different samples, we calculated the total frequency of ATAC insertions per nucleotide. The number of ATAC-Seq insertions located in a 200-bp window centering at TF motifs was then normalized by dividing by the total number of insertions in the flanking 300 bp window. For each sample, we averaged the normalized insertions centered on the TF motif across all footprint sites. To quantitatively compare footprints among datasets, we calculated footprint occupancy score (FOS), following

$$\text{FOS} = \min\left(-\log_2\frac{N_C + 1}{N_L + 1}, -\log_2\frac{N_C + 1}{N_R + 1}\right)$$

where $N_C$ indicates the number of insertions in central region of TF motif. The size of the central region is equal to the length of TF motif. $N_L$ and $N_R$ are separately 1/3 of the numbers of the insertions in the left and right flanking regions of TF motif, as an interval 3 times greater than the size of central region was chosen as the size of flanking region.

**Gene expression analysis**. Using RNA-Seq, we measured gene expression of the retina and RPE from AMD2 left and right eyes. Using Tophat and Cufflinks with default parameters[56], raw data were mapped to the hg19 genome assembly, and gene expression FPKM (Fragments Per Kilobase Of Exon Per Million) were obtained. In the study, we also used normal, pre-AMD and dry AMD samples from published expression dataset (GEO accession number: GSE29801)[19]. If multiple ATAC-Seq peaks were associated with the same gene, we selected the highest ATAC-Seq peak among the gene-proximal peaks to associate with gene expression. To compare differential chromatin accessibility and differential mRNA expression, we only included genes with the average expression more than 1 FPKM in the left and right eyes of patient AMD2.

**Analysis of GWAS SNPs in DARs**. To study the association of DARs with SNPs, we downloaded all SNPs (single nucleotide polymorphisms) from the latest genome-wide association studies (GWAS) performed in AMD[5]. GWAS SNPs were assigned to corresponding ATAC-Seq peaks. We chose the minimum P value of SNPs in each peak region as the significance of ATAC-Seq peak associated with GWAS SNPs. For ATAC-Seq peaks without any associated SNP, the P value of the peak was set to 1. Similar analyzes were performed for DARs, non-DARs, and non-peaks. Flanking regions with the same size of peaks were chose as non-peaks. We then calculated the proportion of DARs, non-DARs, and non-peaks whose P values were less than a fixed threshold.

**Cell type-specific genes associated with DARs**. We collected seven sets of genes which are specifically expressed in rods, cones, horizontal cells, bipolar cells, Müller glia, amacrine cells and ganglion cells, which were assembled from previously published data from mouse[57–63]. Using information downloaded from MGI database (http://www.informatics.jax.org/), we mapped these genes to their corresponding human orthologues. We then compared cell type-specific genes to genes in DARs, and identified the number of cell type-specific genes in each category associated with DARs.

**Human iPSC-derived RPE cells and plasmid transfection**. Human induced pluripotent stem cells (iPSC) were first reprogrammed from a well-characterized fetal fibroblast neonatal cell line[64,65]. The iPSC cell line was then induced to differentiate into RPE cells as previously described[66,67]. Briefly, the iPSC line were maintained on growth factor-reduced Matrigel (BD Biosciences) in mTeSR1 medium (Stem Cell Technologies), in a 10% $CO_2$ and 5% $O_2$ incubator and amplified by clonal propagation using the ROCK pathway inhibitor Blebbistatin (Sigma). For differentiation, iPSC were plated at higher density (25,000 cells per $cm^2$) and maintained in mTeSR1 to form a monolayer and the culture medium was replaced with differentiation medium (DM) for 40–45 days. Differentiating cells were enzymatically dissociated using 0.25% (wt/vol) collagenase IV (Gibco) and resuspended in AccuMAX (Sigma-Aldrich) to make single cell suspension. Cells were re-plated in fresh Matrigel coated plates and maintained in RPE medium [70% DMEM, 30% Ham's F12 nutrient mix, 2% B27-serum-free supplement liquid, 1% antibiotic-antimycotic solution (Invitrogen)] for 2–3 months to form mature RPE monolayers.

Immunostaining for RPE-specific markers were performed using the IntraPrep Permeabilization kit (Beckman Coulter) following the manufacturer's instructions. Primary antibody concentration was 0.046 μg per 1 million cells for mouse anti-RPE65 (1:50, Abcam, ab13826, #401.8B11.3D9) and mouse anti-RLBP1 (1:100, Abcam, ab15051, #B2). Goat anti-mouse conjugated to Alexa 647 (1:1000, Invitrogen, #A28181) was used as a secondary antibody. Mouse (G3A1) mAb IgG1 isotype control (1:50 or 1:100, Cell Signaling, #5415) was included in all flow cytometry experiments and stained cells were analyzed using a C6 flow cytometer (Accuri™). We used the non-specific but species-appropriate isotype control (mAb IgG1) as a negative control to gate the non-specific background signal caused by the primary antibody. This gating strategy enabled us to differentiate non-specific background signal from RPE65- and RLBP1-specific antibody signals.

Plasmids were transfected using DNA-In Stem (2 μL/well) Transfection Reagent (MTI-Global Stem) in mature iPSC-RPE monolayers (>2-month-old) cultured in 24 well plates. 1 μg of plasmid DNA containing empty vector (pCAGIG)[68] or expression vector pCAGIG-HDAC11 were used for transfection. pCAGIG-HDAC11 expression constructs were derived from a pENTR™221 donor plasmid containing a full-length Ultimate ORF™ Entry Clone (Invitrogen) encoding human HDAC11 variant 1 (IOH9974), which was inserted into a variant of pCAGIG in which a Gateway entry cassette had been inserted at the EcoRV site, using LR Clonase™ (Invitrogen). The sequence of the modified entry vector was then confirmed using Sanger sequencing. Plasmid DNA-DNA-In Stem reagent complex is prepared in OptiMEM/RPE medium without antibiotics and incubated for 30 min at RT. Plasmid DNA-DNA-In Stem reagent complex mixture is directly added to the RPE monolayers. Cells were harvested at 72 h after transfection for total protein extraction.

**Cigarette smoke treatment of iPSC-derived RPE cells**. We performed cigarette smoke extract (CSE) treatment in two iPSC-derived RPE cells (IMR90.4 and EP1). The cells were induced into iPSC using viral (IMR90.4) and nonviral (EP1) reprogramming methods, respectively[64–66].

Differentiated RPE cells were plated at 100,000 cells per $cm^2$ on Matrigel-coated plates and allowed to grow for 4 months in RPE medium, consisting of 70% DMEM (Invitrogen), 30% Ham's F-12 Nutrient Mix (catalog no. 11765; Invitrogen), 1× B27 (Invitrogen), and 1× antibiotic-antimycotic (Invitrogen). When cells were fully differentiated after 4 months, they were exposed to Cigarette Smoke Extract (500ug/ml) (Murty Pharmaceuticals, Lexington, KY). We used a "chronic" CSE exposure protocol where the 500 μg/ml dose was used for 2 hours/ day and then removed for each of 5 days. After 5 days of treatment, chromatin and protein were extracted separately for ATAC-Seq and HDAC11 assay.

**Preparation of cell lysates and Western blotting analysis**. iPSC derived RPE monolayers were rinsed twice with ice-cold PBS and collected by scrapping and lysed with RIPA buffer (20–188, Millipore) supplemented with 1% protease inhibitor cocktail (Sigma-Aldrich). Samples were incubated on ice for 30 minutes and vortexed 2–3 times during the incubation. After that, samples were centrifuged at 13,000 g for 20 min. The supernatants were mixed with 4× protein sample buffer (Invitrogen) plus 5% 2-mercaptoethanol (Sigma-Aldrich) and heated at 100 °C for 10 min.

The Western blotting protocol was described previously[69]. Briefly, each sample was loaded onto a 4–12% Bis-Tris Nu-PAGE gel (Invitrogen) and transferred to nitrocellulose membranes. Blots were incubated with anti-HDAC11 antibody (1:1000, rabbit IgG from Thermofisher, PA5-11250) or β-tubulin antibody (1:1000, Cell signaling, #2128) overnight at 4 °C followed by incubation with IRDye infrared dye-conjugated goat-anti-rabbit secondary antibodies at 1:15,000–1:20,000 (LI-COR Biosciences, Lincoln, NE) for 1 h at room temperature. Blots were visualized using using the LI-COR imager system Odessey Classic (LICOR).

Coomassie staining was performed to detect total protein of each sample. After electrophoresis, prefix gel in 50% methanol and 7% acetic acid solution for 15 min. After washing, gels were incubated in GelCode Blue Stain Reagent (Thermo Fisher Scientific™) for 1 h and then destained by washing in water with several changes over 1-2 h. Gels were scanned and the densitometric analysis was done by Quantity One software (Bio-Rad Laboratories).

**Histone extraction**. Histones were isolated with a histone extraction kit (Abcam, ab113476), following the manufacturer's procedure. The protein concentration was determined as described above. Histone Protein (6 μg) was separated by 4–20% Novex Tris-Glycine Gels, and transferred to nitrocellulose membranes. Incubating with primary antibody at 1:1000 (H3K27ac, and Histone 3 (purchased from Abcam) overnight at 4 °C. The rest of the procedure is the same as Western blot described above.

**Statistical analysis**. All statistical analyzes were performed in R platform (https://www.R-project.org/). Fisher's exact test, Student's t-test, one-way ANOVA, Pearson's correlation coefficient were used to assess the significance. Fold change, P-values and FDR (false discovery rate) were calculated in analysis.

**Data availability**. The data of ATAC-Seq and RNA-Seq data in this study have been deposited in NCBI's Gene Expression Omnibus (GEO) under accession number GSE99287.

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

## Acknowledgements

This work was supported by NIH grants R01EY024580 (to J.Q.), R01EY023188 (to S.L.M. and J.Q.), R01EY020560 (to S.B.), R01EY027691, Macular Degeneration Foundation (to J.T.H.), R01NS091165 (to S.X.), and RPB/IRRF Catalyst Award for Innovative Research Approaches for AMD (D.S.). We thank Akrit Sodhi, Jeff Mumm, and Albert Jun for insightful discussions.

## Author contributions

J.Q. and S.B. proposed the project. J.W., C.Z., and H.J. conceived the method. C.Z., P.S., S.R.S, P. Z., T.H., J.T.H., S.X., and M.C. conducted experiments. J.W. performed data analysis. J.Q., S.B., J.W., and C.Z. wrote the manuscript. J.Q., S.B., J.W., C.Z., H.J., S.L.M., D.J.Z., J.T.H., and D.S. revised the manuscript. J.Q. and S.B. supervised this work.

## Additional information

**Conflict of interest:** The authors declare no competing interests.

