## [Peer Review File · Nature Communications]

Reviewer #1 (Remarks to the Author):

This study submitted by an outstanding group of scientists in the the field of age-related macular degeneration (AMD) research that there are differences with respect to human ocular tissue type and disease phenotype with respect to decreases in global chromatin accessibility, for which HDAC11 is implicated. Moreover the authors show that retinal pigment epithelium cells (RPE) treated exposed to cigarette smoke mimics epigenetic changes observed in AMD.

The authors correctly point out that there are no suitable animal models to study the leading cause of blindness (AMD) in the aging population. Therefore it is necessary to obtain human eye tissue to uncover disease mechanism for such studies which presents its own sets of challenges. The authors are to be commended for phenotyping the eye tissue postmortem by an experienced AMD retinal specialist(as most studies do not do this) but rely sometimes on self report, medical record only, etc. This helps to ensure that downstream experiments are robust to meaningful and impactful interpretation.

This is a novel and important study, however few things to be addressed:

1. According to the table 1, the results, and methods, 16 eyes from 5 controls and 5 AMD donors were used. It is not clear how many of these eyes were paired ie., from the same donor n/10? Of the ones that were paired how many were dissimilar and/or similar in their phenotype? For example OD was normal but OS was early AMD? There is one example on page 6 (lines 121 -124) where a donor was examined whose AMD status was asymmetrical and the potential of genetic and environmental factors was excluded. It might be helpful if other pairs were examined in this manner including "controls" where both donor eyes were symmetrical. Particularly since this is a study on epigenetics this would help strengthen results and hence conclusions.

2. Since most studies on the advanced form of AMD focus on the wet neovascular subytpe and this study focused on the advanced geographic subtype this should be emphasized in the results and even in the introduction as this is unique. What is being found here at the global chromatin level may be specific to the early and geographic forms of AMD.

3. It is not clear if the choroid was separated from the RPE or this should be written as RPE/choroid particularly for the normal and early forms of AMD.

4. The authors are to be commended for examining whether or not the observed changes in chromatin stability for each DAR are due to AMD associated variants.

5. For the iPSC-derived RPE cells, although RPE-specific markers were employed to examine their integrity-it is not clear if the iPSC-derived RPE cells also subjected to RNASeq? Was their gene expression profile then compared to normal human RPE tissue ? Maybe this has been done previously?

6. Can authors please comment in discussion on how postmortem time of tissue procurement may or may not influence interpretation of the gene expression experiments and analysis?

Reviewer #2 (Remarks to the Author):

The major claims to this paper are 2 fold, firstly, that global decreases in chromatin accessibility occur in RPE in early Age-related macular degeneration (AMD), whereas similar changes are only seen in the retina with advanced disease suggesting that dysfunction in RPE cells drives disease progression. Secondly, treatment of RPE cells by cigarette smoke recapitulates the epigenomic changes seen in AMD with overexpression of HDAC11 being partially responsible for this observation.

Epigenetic studies have been undertaken before on AMD including changes in methylation (Hunter et al, Invest Ophthalmol Vis Sci. 2012 Apr; 53(4): 2089–2105, Wei et al, Cell Rep. 2012 Nov 29;2(5):1151-8) either on donor eyes or from blood. HDACs have also been implicated in retinal diseases.

The main strengths of the study come from observations appearing to indicate differences between controls (no disease), early AMD and late AMD offering a potential insight into a biological phenomenon that has so far not been effectively interrogated in AMD. However, the evidence, while promising, is still limited in that the current study has collected donor eyes from a small number of 5

controls and 5 donors and then both eyes have been used in the majority of cases for subsequent analyses. The authors draw their conclusions based on differences in global profiling on disease versus non disease but do not seem to account for any biases that may exist with the use of both eyes from the same individual. Thus the analysis should be redone taking into account the use of an eye from each unique donor to avoid such bias. Given the small pool of donor eyes, the conclusions drawn would be strengthened through replication using either an independent set of donor samples or at least top genes should be investigated using a second technique to validate the robustness of the reported findings. While the timing from death to retrieval is up to 14 hours, timing variation between samples could have an impact on the data collected and conclusions drawn and so further analyses documenting how chromatin changes with collection time should be undertaken and reassessed in light of this timing. Interestingly in the paper by Hunter et al, they used double the number of donor eyes for their studies and their timing from collection to retrieval was much less. The current study does not appear to cite this study or other studies on the general topic of epigenetics in AMD. This could lead to confusion to the reader as to the novelty of the current paper or whether a better argument could be made as to level of consensus from the available data being presented at different levels of epigenetic studies. It is also unclear how the authors identified tens of thousands of peaks in their comparisons but then defined in some cases less than 10 as being significant – what level of significance was used and what was the “n” in this calculation?

The cigarette smoking assay is the weaker part as a single cell line (EP1) was promoted as the iPSC line of comparison to reaffirm changes expressed with smoking. The study has an n=1 which would not appear appropriate. Also the cell line used would appear to be from an RP patient (although the citation for the cell line is not provided) rather than being an AMD cell line or better still, an iPSC from an AMD patient. Likewise, the 4 month time period which to generate an RPE cell like structure has required multiple passaging. There is thus the notion that the changes observed in the paper may result from passaging thereby leading to alteration in cellular changes rather than from a smoking effect. The authors have not confirmed karyotypic changes have not occurred at this passaging age nor do they appear to have replicated their findings in different clonal colonies or cell line readouts? How relevant is an RP cell line to AMD? How would the effective age of the cells mimic those seen in AMD? How reproducible are the findings? There does not appear to be any confirmatory experiments presented in the paper.

Reviewer #3 (Remarks to the Author):

Comments for the Authors

This paper by Wang et al. characterizes the genome-wide chromatin accessibility landscape in human tissue samples (retina and retinal pigmented epithelium, RPE) obtained from 5 control donors and 5 patients with age-related macular degeneration (AMD). The authors report global decreases in chromatin accessibility in retina and RPE tissue from AMD patients, as well as retina and RPE specific chromatin accessibility signatures, both of which should be of interest to many researchers in the field. The paper also reports similar changes in chromatin accessibility in iPSC-derived RPE upon exposure to cigarette smoke extract or HDAC11 overexpression. Given the difficulty of obtaining sufficient donor tissue and the lack of genome-wide epigenetic data for AMD, this is an important and interesting paper. Moreover, several elegant controls suggest the ATAC-seq results are reliable and reproducible. However, several changes would improve the manuscript, primarily regarding the iPSC-RPE cell culture model, to compliment and improve the data.

Major Comments:

1. While it will be important to determine the mechanism(s) underlying the AMD-induced decrease in chromatin accessibility demonstrated here, the data linking HDAC11 to these changes is not yet sufficiently compelling.
 - a. The paper cites previous microarray data (GSE29801) indicating increased HDAC11, but fails to mention that two other histone deacetylases (HDAC10 and SIRT1) are significantly downregulated in the same data set (Extended Data Table 4). This should be noted in the discussion.
 - b. It would be informative to examine the global level of key histone acetylation marks (for example H3K9ac or H3K27ac) in cigarette smoke treated iPSC-RPE. This could be performed by western blot on the protein samples from Extended Data Figure 5D using commercially available antibodies.
 - c. Extended Data Figure 5D,F: It is not clear from the western blots presented that a meaningful increase in HDAC11 protein level is occurring or being measured in the iPSC-RPE.
 - i. While the HDAC11 western blots indicate a dimer band at >62kDa, HDAC11 has a predicted molecular weight of 39kDa and the antibody data sheet (Abcam ab18973) reports a band of approximately the expected size. It would be important to verify that the >62kDa band observed in the paper indeed represents HDAC11, for example with a different antibody or using RNAi.
 - d. Although cigarette smoke produces a smaller increase in HDAC11 level (~25%) than HDAC11 overexpression (~2.5 fold), the ATAC-Seq data for cigarette smoke is more highly correlated with AMD than that of HDAC11 overexpression (R=0.36 vs R=0.24, respectively). This implies that HDAC11 level plays a minor role in the cigarette smoke induced changes in chromatin accessibility, which should be mentioned.

Minor Comments

2. Introduction (line 48-49): While current knowledge of epigenetic mechanisms in AMD is indeed limited, the introduction should cite relevant previous work on DNA methylation in AMD patients (for example, Hunter et al. 2012 [PMID22410570], Wei et al. 2012 [PMID23177625], and Oliver et al. 2013 [PMID24373284]).
3. Figure 1D (line 102-103): The text states that most AMD samples clearly cluster separately from controls, whereas visual inspection of Figure 1D indicates ~7 AMD samples that are clearly separated from the clusters while many AMD samples are nearly overlapping with controls. This clustering analysis should be summarized quantitatively if possible.
4. Line 404: "Beside of GSEA database..." appears to be a typographical error.
5. Line 483-484: The source of the pCAGIG-HDAC11 plasmid and/or details regarding its construction should be present.
6. Rationale for the dose of cigarette smoke extract used (500ug/ml) should be given, as it is 5-10X higher than that used in several other publications.
7. Extended Data Figure 5C,E: Clarification on the meaning of symbols on the graphs (+, x, *) should be added to the figure legend.

We are grateful to the three reviewers for their overall positive and constructive comments. We have provided new data and new analyses to address the concerns raised by the reviewers. Please find below our detailed point-by-point response.

Reviewer #1 (Remarks to the Author):

This study submitted by an outstanding group of scientists in the the field of age-related macular degeneration (AMD) research that there are differences with respect to human ocular tissue type and disease phenotype with respect to decreases in global chromatin accessibility, for which HDAC11 is implicated. Moreover the authors show that retinal pigment epithelium cells (RPE) treated exposed to cigarette smoke mimics epigenetic changes observed in AMD.

The authors correctly point out that there are no suitable animal models to study the leading cause of blindness (AMD) in the aging population. Therefore it is necessary to obtain human eye tissue to uncover disease mechanism for such studies which presents its own sets of challenges. The authors are to be commended for phenotyping the eye tissue postmortem by an experienced AMD retinal specialist (as most studies do not do this) but rely sometimes on self report, medical record only, etc. This helps to ensure that downstream experiments are robust to meaningful and impactful interpretation.

Response: Thank the reviewer for the positive comments and the appreciation of our careful experimental design.

This is a novel and important study, however few things to be addressed:

1. According to the table 1, the results, and methods, 16 eyes from 5 controls and 5 AMD donors were used. It is not clear how many of these eyes were paired ie., from the same donor n/10? Of the ones that were paired how many were dissimilar and/or similar in their phenotype? For example OD was normal but OS was early AMD? There is one example on page 6 (lines 121 -124) where a donor was examined whose AMD status was asymmetrical and the potential of genetic and

environmental factors was excluded. It might be helpful if other pairs were examined in this manner including "controls" where both donor eyes were symmetrical. Particularly since this is a study on epigenetics this would help strengthen results and hence conclusions.

Response: We sometimes received only one high quality eye globe from one donor. Among the 10 donors, we have 6 paired-eyes. Among the 6 paired-eyes, only one pair of eyes from the same donor was dissimilar whose OD was early AMD and OS was late AMD (see Figure 2c). Besides examining this donor whose AMD status was asymmetrical, we indeed have already performed the analysis of changes in chromatin accessibility for both donor eyes that had symmetrical disease severity. These analyses served as "negative controls". A few examples were presented in the manuscript (see Figure 2e and Extended Data Fig. 2a for the retina and Figure 3c and Extended Data Fig. 2d for the RPE). We have emphasized the analysis of the "controls" in the revised manuscript (page 7, paragraph 1 and Extended Data Table 1).

2. Since most studies on the advanced form of AMD focus on the wet neovascular subtype and this study focused on the advanced geographic subtype this should be emphasized in the results and even in the introduction as this is unique. What is being found here at the global chromatin level may be specific to the early and geographic forms of AMD.

Response: Thanks for your suggestion. We agree that this distinction is one of the novelties of our study. We have emphasized this unique aspect of the potential impact of global chromatin changes in both early and the advanced geographic subtype of AMD by adding, "Here, we focused on less reported but more prevalent atrophic ("dry") AMD." and "Most studies have focused on neovascular AMD. In contrast, our study focused on changes at the highly prevalent early stage and the less reported late, atrophic stage of AMD." in the Introduction on page 3, and in the Discussions on page 13.

3. It is not clear if the choroid was separated from the RPE or this should be written as RPE/choroid particularly for the normal and early forms of AMD.

Response: We added more details on our method in the Methods. "Under direct visualization with a dissecting microscope, the RPE was mechanically separated from the choroid." on page 16.

To quantitatively assess for potential contamination from the choroid, we examined the expression of choroid-specific genes in RPE samples using RNA-seq. Compared to RPE-specific genes (e.g. PMEL, RPE65 and BEST1), the expression of choroid-specific genes (NG-2, PV-1, CD31) was minimal (inserted figure A), suggesting the high purity of RPE samples. We include this new analysis in the revised version of manuscript (Page 4, paragraph 2, and Extended data Figure 1a).

Figure A. Expression of RPE- and choroid-specific genes in the RPE samples. The averaged expression values are from RPE tissues of left and right eyes of one donor measured by RNA-seq. The error-bar represents the standard error of the mean.

4. The authors are to be commended for examining whether or not the observed changes in chromatin stability for each DAR are due to AMD associated variants.

Response: Thank you. In fact, we examined whether the observed changes in chromatin accessibility for each DAR are due to AMD associated variants in the section, “Epigenetic changes generally occur independently of AMD risk-associated genetic variants” (see page 11, paragraph 1).

5. For the iPSC-derived RPE cells, although RPE-specific markers were employed to examine their integrity-it is not clear if the iPSC-derived RPE cells also subjected to RNaseq? Was their gene expression profile then compared to normal human RPE tissue? Maybe this has been done previously?

Response: As the reviewer points out, we already showed the expression of RPE-specific markers in the iPSC-derived RPE cells (Extended data Fig. 5a). To further confirm the integrity of these cells, we compared the overall expression profiles between the iPSC-derived RPE cells and normal human RPE tissue. We performed RNA-seq for the iPSC-derived RPE cells, and compared it with the expression of normal human RPE tissue obtained from a published dataset (**Ref. 1**). The correlation coefficient of the two datasets is 0.81, suggesting the overall transcriptome of iPSC-derived RPE cells is very similar to that of normal human RPE cells (see insert Figure B). We would like to point out that we also compared the chromatin accessibility profiles between iPSC-derived RPE cells and human RPE cells (see Extended data Fig. 5c), demonstrating that the iPSC-derived RPE cells are similar to normal human RPE cells not only at transcriptome level, but also at epigenomics level. We included the new analysis in the revised version (Page 12, paragraph 1, and Extended data Fig. 5b).

Figure B. Comparison of expression profiles between normal RPE tissue (n = 8) and iPSC-derived RPE (n = 3).

6. Can authors please comment in discussion on how postmortem time of tissue procurement may or may not influence interpretation of the gene expression experiments and analysis?

Response: In our linear regression model, we took into account the potential confounding factors including procurement interval. Our analysis indicated that the postmortem time of tissue procurement was positively correlated with reduced chromatin accessibility. In other words, longer procurement time leads to greater reduced chromatin accessibility. The control donors had longer procurement time than AMD donors (see Table 1 and Extended Data Table 1). Therefore, the actual chromatin reduction could

be even more significant that we report herein. We added more discussion on this aspect (see page 9, paragraph 1).

Reviewer #2 (Remarks to the Author):

1. The major claims to this paper are 2 fold, firstly, that global decreases in chromatin accessibility occur in RPE in early Age-related macular degeneration (AMD), whereas similar changes are only seen in the retina with advanced disease suggesting that dysfunction in RPE cells drives disease progression. Secondly, treatment of RPE cells by cigarette smoke recapitulates the epigenomic changes seen in AMD with overexpression of HDAC11 being partially responsible for this observation.

Epigenetic studies have been undertaken before on AMD including changes in methylation (Hunter et al, Invest Ophthalmol Vis Sci. 2012 Apr; 53(4): 2089–2105, Wei et al, Cell Rep. 2012 Nov 29;2(5):1151-8) either on donor eyes or from blood. HDACs have also been implicated in retinal diseases.

Response: We thank the reviewer for pointing out the relevant references. While these two papers examined DNA methylation in AMD, our work focused on chromatin accessibility, and to our understanding, our study is the first comprehensive analysis of the chromatin accessibility changes in AMD. We believe that chromatin accessibility is quite distinct from methylation changes that Hunter et al, others, including our group have already reported, and importantly, this type of change has never been associated with AMD. In our revised manuscript, we cited these two papers and our own study on DNA methylation changes in AMD (See Page 3, “Several groups have reported that DNA methylation changes in individual genes may be associated with AMD⁹⁻¹²”).

2. The main strengths of the study come from observations appearing to indicate differences between controls (no disease), early AMD and late AMD offering a potential insight into a biological phenomenon that has so far not been effectively interrogated in AMD. However, the evidence, while promising, is still limited in that the current study has collected donor eyes from a small number of 5 controls and 5 donors and then both eyes have been used in the majority of

cases for subsequent analyses. The authors draw their conclusions based on differences in global profiling on disease versus non disease but do not seem to account for any biases that may exist with the use of both eyes from the same individual. Thus the analysis should be redone taking into account the use of an eye from each unique donor to avoid such bias. Given the small pool of donor eyes, the conclusions drawn would be strengthened through replication using either an independent set of donor samples or at least top genes should be investigated using a second technique to validate the robustness of the reported findings. While the timing from death to retrieval is up to 14 hours, timing variation between samples could have an impact on the data collected and conclusions drawn and so further analyses documenting how chromatin changes with collection time should be undertaken and reassessed in light of this timing. Interestingly in the paper by Hunter et al, they used double the number of donor eyes for their studies and their timing from collection to retrieval was much less. The current study does not appear to cite this study or other studies on the general topic of epigenetics in AMD. This could lead to confusion to the reader as to the novelty of the current paper or whether a better argument could be made as to level of consensus from the available data being presented at different levels of epigenetic studies. It is also unclear how the authors identified tens of thousands of peaks in their comparisons but then defined in some cases less than 10 as being significant – what level of significance was used and what was the “n” in this calculation?

Response: Here the reviewer has raised several technical concerns. Please find below our detailed response to each concern.

Sample size. The reviewer questioned the statistical significance of our conclusions due to a relatively small sample size, and indicated the sample size in our study is small because “Interestingly in the paper by Hunter et al, they used double the number of donor eyes for their studies”. However, there is no “one size fits all” regarding sample size. As you are aware, if the biological effect is small, a large sample size is necessary to determine a meaningful effect. On the other hand, if the biological effect is large, a small sample size is sufficient to draw a meaningful conclusion. Based on our own extensive experience studying DNA methylation, we agree that the changes in DNA methylation are often very small, and for such studies large sample sizes are important. For example, in one of our published studies (**Ref. 2**), we were unable to find any genome-wide statistically significant differential methylation sites between AMD and control (100 AMD vs 99 controls in blood, 9 AMD vs 9 controls in retina). Therefore, it is not surprising to us that Hunter et al needed a large sample size to identify a

difference in methylation sites in the retina. In stark contrast, with the sample size we presented, we observed a large and consistent change in chromatin accessibility between AMD and controls using multiple different analyses. First, when we compared all AMD to control samples, 92.3% and 91.6% peaks showed decreased intensity in the retina and RPE, respectively (Figure 2b and Figure 3a). Second, in a unique case, one donor had asymmetric degrees of AMD, one eye with early and the other with late stage AMD (Figure 2c). The comparison of these two eyes from the same donor also showed that 76.7% and 87.1% of peaks have decreased intensity in the retina and RPE, respectively (Figure 2d and 3b), suggesting that the effect size of chromatin accessibility changes in AMD is indeed large. This comparison provided an orthogonal analysis because the contributions of potentially confounding genetic and environmental influences that might complicate the analysis of epigenetic changes associated with AMD progression are excluded. Third, we analyzed the chromatin changes that occurred in both the early and late stages of AMD. Based on this analysis, 939 and 5,458 peaks were significantly decreased in the retina and RPE, respectively (Figure 3d). The three independent analyses revealed a different number of decreased ATAC-seq peaks. However, it is clear from these data that there is a widespread reduction in chromatin accessibility associated with AMD. Our findings are highly statistically significant and illustrate the scientific rigor of our conclusions. Therefore, we disagree that a larger sample size is necessary to support our conclusions.

The reviewer also suggested a new method to confirm our observation by “using an eye from each unique donor to avoid bias”. Following the reviewer’s suggestion, we performed a new analysis. Instead of including two eyes from the same donor, we used one eye from each of 10 donors. Based on the new analysis, we found that 88.6% and 86.2% of peaks have decreased ATAC-seq peaks in the retina and RPE, respectively (see inserted Figure C). Furthermore, we modified the linear regression model and added one variable to account for the donor ID. The modified model takes into account any biases that may exist with by using both eyes from the same individual. The new model showed that 43,392 (55.1%) peaks in the retina and 45,092 (91.6%) peaks in the RPE have decreased intensity with a false discovery rate of 0.05 (see inserted Figure D). Our new analysis shows a robust difference between controls and AMD eyes, and that our conclusions remain the same. We also added this information to the Results (Page 7, paragraph 2, and Extended data Fig. 2g and 2h).

Figure C. Changes in chromatin accessibility between AMD and normal samples. Only one retina (top panel) or RPE (bottom panel) sample from each of 10 donors was used. Blue line indicates average fold changes of peaks. The percentage of reduced peaks is shown under the density curve.

Figure D. The density of the stage coefficients in the modified linear regression model.

We also point out that while a replication cohort has been used in genome-wide association studies (GWAS), it is not a common practice in epigenomics studies. Although varying number of samples have been used depending upon the specific biological question, epigenomics studies typically do not employ a replication cohort, including the two DNA methylation papers mentioned by the reviewer and other

published works (**Refs. 3 and 4**). For example, in the study from Schultz MD, et al, DNA methylation profiles were determined in various tissue types from 4 individuals and the DNA methylation patterns were compared among these tissues. A replication cohort was not used because the biological effect was significant. One technical reason for such an experimental design in epigenomics studies is the difficulty with obtaining relevant tissue for these studies. For instance, it took us 2 years to get the fresh, high quality eye globes from 10 donors that were suitable for use in our study. We added a paragraph in Discussion for this issue (See Page 14, paragraph 3).

Procurement timing variation. Our linear regression model included procurement timing variation (page 9, paragraph 1). Our analysis and results indicate that the procurement interval unlikely leads to the decreased peak intensity in AMD samples that was observed in this study, given that the procurement interval of AMD samples is shorter than normal samples (Table 1). We added more discussion on Page 9.

Peak number difference. When we included all samples to compare the chromatin accessibility between AMD and controls, we identified tens of thousands of peaks using a linear regression model (page 9). However, when we separated the AMD samples into early and late stage disease, and compared early AMD vs control, and late AMD vs early AMD, we identified different numbers of differential accessibility regions (DARs) using an exact test analogous to Fisher's exact test through EdgeR software. In the retina, we observed 939 DARs when comparing late vs early AMD, and only 3 DARs when comparing early AMD vs. controls. In contrast, in the RPE, we identified 5,458 DARs when comparing early AMD vs. controls, and only 2 DARs when comparing late vs. early AMD. The huge difference of DARs from our comparisons of the retina and RPE suggest that AMD is likely initiated from the RPE. We have chosen $FDR < 0.05$ as significant level and described the number of samples in each comparison in the main text on page 8, paragraph 2 and paragraph 3.

Citation of previous DNA methylation studies. In the revised manuscript, we cited the two studies mentioned by the reviewer and our own DNA methylation study on AMD (see introduction on page 3).

3. The cigarette smoking assay is the weaker part as a single cell line (EP1) was promoted as the iPSC line of comparison to reaffirm changes expressed with smoking. The study has an n=1 which would not appear appropriate. Also the cell line used would appear to be from an RP patient (although the citation for the cell line is not provided) rather than being an AMD cell line or better

still, an iPSC from an AMD patient. Likewise, the 4 month time period which to generate an RPE cell like structure has required multiple passaging. There is thus the notion that the changes observed in the paper may result from passaging thereby leading to alteration in cellular changes rather than from a smoking effect. The authors have not confirmed karyotypic changes have not occurred at this passaging age nor do they appear to have replicated their findings in different clonal colonies or cell line readouts? How relevant is an RP cell line to AMD? How would the effective age of the cells mimic those seen in AMD? How reproducible are the findings? There does not appear to be any confirmatory experiments presented in the paper.

Response: The reviewer raised several technical concerns. We address each concern below.

Cell origin iPSC-RPE line (EPI) is NOT derived from the RP patients. Instead, we obtained it from a well-characterized fetal fibroblast (IMR90) normal cell line from ATCC. The iPSC-RPE cells were induced to become RPE cells using the protocol established by the Zack lab (Ref. 5). We added more information in the revised manuscript (See Methods on Page 23, paragraph 3).

Multiple passages We agree with the reviewer's concern that multiple passages can influence the phenotype. In general, significant changes can be induced after passage 4. In our experiment, iPSC cells were differentiated to RPE cells and then passaged once. Thus, at passage 2 as RPE cells, they were grown for 4 months. We know from our work that they develop characteristics of terminal differentiation, such as a cobblestone cell shape, melanin pigmentation, and the production of visual cycle proteins such as RPE65 and LRAT. We have also compared the genome-wide expression profile of the iPSC-derived RPE cell with normal human RPE cells, and find that the expression profiles are very similar (see reviewer #1's point 5. Please see our new analysis on Page 12). Furthermore, the chromatin accessibility changes were obtained by comparing the iPSC-derived cells before and after CSE treatment. Therefore, the observed changes in chromatin accessibility were induced by CSE, rather than from passaging. We understand the reviewer concerns regarding possible genotoxic effect of cigarette smoke extract (CSE) treatment in RPE monolayers. We have analyzed the karyotype of hiPSC-RPE cells and they are apparently normal cells (see Figure E below). We added the new results in Extended Data Fig. 5d.

Figure E. Karyotype analysis of hiPSC-RPE cells.

Biological replicates. We indeed have two biological replicates for this analysis. The results shown in Figure 4e and 4f were the average of the replicates. The comparison between the two replicates showed great reproducibility (see inserted Figure F). Therefore, we are confident with the observed changes in chromatin accessibility induced by the cigarette smoke treatment. We included the new results in the revised manuscript (Page 12, paragraph 2, and Extended Data Fig. 5e), and make it clear that we have replicates in our analysis.

Figure F. Comparison of chromatin accessibility in two replicates of cigarette smoke-treated RPE monolayers.

Reviewer #3 (Remarks to the Author):

Wang et al. A widespread decrease of chromatin accessibility in age-related macular degeneration

This paper by Wang et al. characterizes the genome-wide chromatin accessibility landscape in human tissue samples (retina and retinal pigmented epithelium, RPE) obtained from 5 control donors and 5 patients with age-related macular degeneration (AMD). The authors report global decreases in chromatin accessibility in retina and RPE tissue from AMD patients, as well as retina and RPE specific chromatin accessibility signatures, both of which should be of interest to many researchers in the field. The paper also reports similar changes in chromatin accessibility in iPSC-derived RPE upon exposure to cigarette smoke extract or HDAC11 overexpression. Given the difficulty of obtaining sufficient donor tissue and the lack of genome-wide epigenetic data for AMD, this is an important and interesting paper. Moreover, several elegant controls suggest the ATAC-seq results are reliable and reproducible. However, several changes would improve the manuscript, primarily regarding the iPSC-RPE cell culture model, to compliment and improve the data.

Response: We thank the reviewer for the positive comments.

Major Comments:

1. While it will be important to determine the mechanism(s) underlying the AMD-induced decrease in chromatin accessibility demonstrated here, the data linking HDAC11 to these changes is not yet sufficiently compelling.

a. The paper cites previous microarray data (GSE29801) indicating increased HDAC11, but fails to mention that two other histone deacetylases (HDAC10 and SIRT1) are significantly downregulated in the same data set (Extended Data Table 4). This should be noted in the discussion.

Response: We thank the reviewer for the suggestion. Since HDACs tend to lead to chromatin accessibility reduction, we searched for overexpressed HDACs instead of downregulated HDACs. In the revised manuscript, we mentioned these two histone deacetylases (HDAC10 and SIRT1) on page 13.

b. It would be informative to examine the global level of key histone acetylation marks (for example H3K9ac or H3K27ac) in cigarette smoke treated iPSC-RPE. This could be performed by western blot on the protein samples from Extended Data Figure 5D using commercially available antibodies.

Response: We thank the reviewer for the suggestion. We examined the global level of histone acetylation marks H3K9ac and H3K27ac. We observed that H3K27ac level decreased with cigarette smoke treatment (see Figure G below), consistent with the observation of increased level of HDAC11. The level of H3K9ac didn't show significant change, indicating that the histone mark might not be the target of HDAC11. We included the new results in Extended Data Fig. 6b.

Figure G. Change of H3K27ac under cigarette smoke treatment.

c. Extended Data Figure 5D, F: It is not clear from the western blots presented that a meaningful increase in HDAC11 protein level is occurring or being measured in the iPSC-RPE.

Response: We performed the experiment again using another HDAC11 antibody (see the next point). The increase in HDAC11 is more significant. More importantly, as suggested by the reviewer, we also examined the global level of H3K27ac using western blot and found that the level of H3K27ac was reduced in cigarette smoke treatment, confirming the observation of HDAC11 increase.

i. While the HDAC11 western blots indicate a dimer band at >62kDa, HDAC11 has a predicted molecular weight of 39kDa and the antibody data sheet (Abcam ab18973) reports a band of approximately the expected size. It would be important to verify that the >62kDa band observed in the paper indeed represents HDAC11, for example with a different antibody or using RNAi.

Response: We tested a few HDAC11 antibodies and found that one antibody from Thermofisher yielded the bands of expected size. We repeated the western blots in both cigarette smoke treatment and HDAC11 overexpression experiments (Extended Data Fig. 6a and 6c). The conclusion still holds with the new antibody (see Page 13).

d. Although cigarette smoke produces a smaller increase in HDAC11 level (~25%) than HDAC11 overexpression (~2.5 fold), the ATAC-Seq data for cigarette smoke is more highly correlated with AMD than that of HDAC11 overexpression (R=0.36 vs R=0.24, respectively). This implies that HDAC11 level plays a minor role in the cigarette smoke induced changes in chromatin accessibility, which should be mentioned.

Response: Thanks for this great suggestion. We acknowledge that cigarette smoke extract induced more chromatin accessibility changes than HDAC11 overexpression, and that there are other possible causes for the chromatin accessibility changes. We added more relevant discussion of HDAC11 in the revised manuscript (Please see Page 15).

Minor Comments

2. Introduction (line 48-49): While current knowledge of epigenetic mechanisms in AMD is indeed limited, the introduction should cite relevant previous work on DNA methylation in AMD patients (for example, Hunter et al. 2012 [PMID22410570], Wei et al. 2012 [PMID23177625], and Oliver et al. 2013 [PMID24373284]).

Response: According to reviewer's suggestion, we cited the relevant previous work on DNA methylation in AMD patients on page 3.

3. Figure 1D (line 102-103): The text states that most AMD samples clearly cluster separately from controls, whereas visual inspection of Figure 1D indicates ~7 AMD samples that are clearly separated from the clusters while many AMD samples are nearly overlapping with controls. This clustering analysis should be summarized quantitatively if possible.

Response: We thank the reviewer for the suggestion. We performed a new analysis (Extended Data Fig. 1e) and included the new results on page 6, paragraph 1.

4. Line 404: “Beside of GSEA database...” appears to be a typographical error.

Response: We corrected this typographical error to “Besides GSEA database...” on page 21.

5. Line 483-484: The source of the pCAGIG-HDAC11 plasmid and/or details regarding its construction should be present.

Response: We provided more information regarding the construction. We added the following in the revised version. “pCAGIG-HDAC11 expression constructs were derived from a pENTRTM221 donor plasmid containing a full-length Ultimate ORFTM Entry Clone (Invitrogen) encoding human HDAC11 variant 1 (IOH9974), which was inserted into a variant of pCAGIG (Ref. 6) in which a Gateway entry cassette had been inserted at the EcoRV site, using LR ClonaseTM (Invitrogen).” on page 25, paragraph 3.

6. Rationale for the dose of cigarette smoke extract used (500ug/ml) should be given, as it is 5-10X higher than that used in several other publications.

Response: As mentioned in the Methods, we used a “chronic” CSE exposure protocol where the 500 ug/ml dose was used for 2 hours/day and then removed for each of 5 days. We have documented that this protocol does not induce cell death as can happen if this higher dose is used over the more standard 24 hour exposure period, as the reviewer alludes to protocols from several publications. Please see Page 25 for modification.

7. Extended Data Figure 5C, E: Clarification on the meaning of symbols on the graphs (+, x, *) should be added to the figure legend.

Response: We clarified the symbols on Extended Data Fig. 5c and 5e. It was added to the figure legend in the extended data. In the revised manuscript, Extended Data Fig. 5c and 5e were replaced by Extended Data Fig. 5f and 6a, respectively.

References

1. Li M, et al. *Comprehensive analysis of gene expression in human retina and supporting tissues. Human molecular genetics* 2014, 23, 4001-4014
2. Oliver VF, et al. *Differential DNA methylation identified in the blood and retina of AMD patients. Epigenetics* 2015, 10, 698-707.
3. Irizarry RA, et al. *The human colon cancer methylome shows similar hypo- and hypermethylation at conserved tissue-specific CpG island shores. Nature genetics* 2009, 41, 178-186.
4. Schultz MD, et al. *Human body epigenome maps reveal noncanonical DNA methylation variation. Nature* 2015, 523, 212.
5. Maruotti, J. et al. *Small-molecule-directed, efficient generation of retinal pigment epithelium from human pluripotent stem cells. Proc Natl Acad Sci USA* 2015, 112, 10950-5.
6. Matsuda T, Cepko CL. *Electroporation and RNA interference in the rodent retina in vivo and in vitro. Proceedings of the National Academy of Sciences* 2004, 101, 16-22.

Reviewer #1 (Remarks to the Author):

Overall the authors have done a comprehensive and in-depth job of addressing the reviewers' concerns.

Reviewer #2 (Remarks to the Author):

Most studies have focused on neovascular AMD. In contrast, our study focused on changes at the highly prevalent early stage and the less reported late, atrophic stage of AMD.

Reviewer reply: Please provide a description of the eyes as to date there is no indication of how many eyes were "early" and how many were "atrophic"

For the iPSC-derived RPE cells, although RPE-specific markers were employed to examine their integrity-it is not clear if the iPSC-derived RPE cells also subjected to RNASeq? Was their gene expression profile then compared to normal human RPE tissue? Maybe this has been done previously?

Reviewer reply: Text states "Comparison of two biological replicates in cigarette smoke-treated iPSC-derived RPE" but figure B in the author response indicates "iPSC-derived RPE(n = 3)" - clarify discrepancy.

Methylation studies.

Reviewer reply: thank you for inclusion and clarifying in the text

Donor eye tissue

Reviewer reply: thankyou for including each eye analysis and clarifying details

“...replication ...is not a common practice in epigenomics studies. Although varying number of samples have been used depending upon the specific biological question”

Reviewer response: Replication can be undertaken in different ways. In the paper by Lai, replication of initial findings was undertaken in sib pairs and subsequently in sporadic cases. The Schultz paper compared data from 18 tissue types (duplicate and triplicate samples) together with transcriptome and genomic data. The Irizarry paper compared 5 colon, 5 controls and 5 brain, liver and spleen samples.

Smoking section. iPSC-RPE line (EP1) is NOT derived from the RP patients. Instead, we obtained it from a well-characterized fetal fibroblast (IMR90) normal cell line from ATCC.

Reviewer response: Only one cell line with duplicates has been tested. Typically several cell lines rather than duplicates of the same cell line are required as evidence.

Genotoxic response and Karyotyping

Reviewer response: The karyotype shown is at a very coarse level that would only reveal major genetic alterations. The authors should undertake a more detailed inspection such as through the use of SKY or through high density SNP genotyping to indicate that no smaller genetic changes have occurred.

Reviewer #3 (Remarks to the Author):

The authors have addressed my concerns.

Reviewer #4 (Remarks to the Author):

In this paper, Wang et al. performed ATAC-seq analysis between AMD patient samples and controls. Based on decreased ATAC-seq signals in AMD patient samples, Wang et al. concluded that the global chromatin accessibility is impaired in AMD patients. Mechanistically, the authors demonstrated that cigarette smoke extract treatment of RPE cells decreased ATAC-seq signal in iPS-cell-derived RPE cells. Overexpression of HDAC11 also lead to decreased ATAC-seq signal.

There are several strengths in this paper. First of all, human tissues in both disease and normal conditions were used to perform ATAC-seq. Those tissues are not easily accessible, and the data generated from these precious samples are informative to the general audience. Second, the global reduction of ATAC-seq signal is quite intriguing. Third, visualization of the data analysis was beautifully represented in many figures.

Despite the efforts to generate and analyze the data, the conclusions from the analysis need to be tuned down. The mechanisms suggested by the paper are not strongly supported by the data. Major concerns for the papers are listed as follows:

1) decreased “global” ATAC-seq signal alone is not sufficient to demonstrate that the global chromatin accessibility is decreased. Since ATAC-seq relies on penetration of Tn5 transposase into the nuclei of the living cells, the permeability of the cell can directly influence the signal. In other words, the reduction of ATAC-seq signals in all three conditions (AMD, cigarette-extract treated RPE, and HDAC11 overexpression) could be explained alternatively by reduction of permeability. In addition, if global chromatin accessibility is indeed reduced in these conditions, one would predict that global gene expression would be reduced. However the authors clearly stated on page 14 that “no global decrease in RNA expression in late-stage AMD was observed”. Therefore this “global reduction in chromatin accessibility” could be a result of technical artifact, or it could have no direct causal relationship to gene expression.

2) For “Chronic” CSE treatment, The authors used 500ug/mL only for only 5 days at 2hours/day. Although CSE treatment has been reported in the literature for other cell types such as T cells and keratinocytes, RPE may have different response to this treatment. Did RPE cells have stress response? Did the proliferation rate of RPE cell change? All those factors could potentially affect ATAC-seq results, without directly impact chromatin accessibility during the interphase.

3) HDAC11 is generally considered as a class IV HDAC. It can shuttle between the cytoplasm and the nucleus. In RPE cells, is HDAC 11 predominantly localized to the nucleus? With HDAC11 overexpression, a logical experiment is to show that H3K27Ac level is altered. However this result was not included.

To convincingly conclude that the “global” chromatin accessibility is decreased in AMD, additional evidence using alternative approaches should be included. For example, could staining of H3K27Ac be performed for tissue sections? Or could western be performed (if sufficient material available) to compare H3K27Ac levels?

Minor comments: Additional details should be included for the figures

1. Figure 1b, please consider adding the data of house keeping genes. Is the accessibility for house keeping genes altered in AMD?

2. Figure 1c, the scale bar needs a label such as "ATAC-seq signal intensity".

2 Figure 2a upper panel: is this from three representative samples, or is this an aggregation of all data for each category?

3 Figure 4 c, f, h: scale bars should be included to show the meanings of the colors.

Reviewer #1 (Remarks to the Author):

Overall the authors have done a comprehensive and in-depth job of addressing the reviewers' concerns.

=====

Reviewer #2 (Remarks to the Author):

Most studies have focused on neovascular AMD. In contrast, our study focused on changes at the highly prevalent early stage and the less reported late, atrophic stage of AMD.

Reviewer reply: Please provide a description of the eyes as to date there is no indication of how many eyes were “early” and how many were “atrophic”.

RESPONSE: The requested information is now provided in page 4, paragraph 2.

For the iPSC-derived RPE cells, although RPE-specific markers were employed to examine their integrity-it is not clear if the iPSC-derived RPE cells also subjected to RNASeq? Was their gene expression profile then compared to normal human RPE tissue? Maybe this has been done previously?

Reviewer reply: Text states “Comparison of two biological replicates in cigarette smoke-treated iPSC-derived RPE” but figure B in the author response indicates “iPSC-derived RPE(n = 3)” - clarify discrepancy.

RESPONSE: We performed three (n=3) biological replicates for the **RNA-Seq** analysis on the iPSC-derived RPE cells, and two biological replicates for the **ATAC-Seq** analysis on the cigarette smoke treatment.

Methylation studies.

Reviewer reply: thank you for inclusion and clarifying in the text

Donor eye tissue

Reviewer reply: thank you for including each eye analysis and clarifying details

RESPONSE: Thank you.

“...replication ...is not a common practice in epigenomics studies. Although varying number of samples have been used depending upon the specific biological question”

Reviewer response: Replication can be undertaken in different ways. In the paper by Lai, replication of initial findings was undertaken in sib pairs and subsequently in sporadic cases. The Schultz paper compared data from 18 tissue types (duplicate and triplicate samples) together with transcriptome and genomic data. The Irizarry paper compared 5 colon, 5 controls and 5 brain, liver and spleen samples.

RESPONSE: We agree with the Reviewer that “replication can be undertaken in different ways”. We indeed performed several analyses to demonstrate the scientific rigor of our findings. First, *both* retina and RPE from AMD and controls showed significant differences in chromatin accessibility. As suggested by reviewer, we repeated the analysis with only one eye from each donor and the results also confirmed our finding. Second, this observation was confirmed by an orthogonal comparison of *two eyes with asymmetric degrees of severity of AMD from the same donor*. Third, cigarette smoke treatment reproduced these changes. Fourth, overexpression of HDAC11 reduced the changes in chromatin accessibility, which partially recapitulates the changes in AMD patients and cigarette smoke treatment. All of these independent findings suggest that our observation is reliable and reproducible.

Smoking section. iPSC-RPE line (EPI) is NOT derived from the RP patients. Instead, we obtained it from a well-characterized fetal fibroblast (IMR90) normal cell line from ATCC.

Reviewer response: Only one cell line with duplicates has been tested. Typically several cell lines rather than duplicates of the same cell line are required as evidence.

RESPONSE: As suggested by the Reviewer, we performed cigarette smoke treatment in one additional cell line. The cell line was generated from fetal lung fibroblasts by viral transduction of a series of key transcription factors such as OCT4, SOX2, NAOG and LIN28 (WiCell Research Institute, Madison, WI). The iPSC cells were then differentiated to RPE cells. The RPE cell line was well characterized in our previous work^{1,2}.

By comparison of chromatin accessibility in control and treated cells, we confirmed that cigarette smoke treatment decreased the chromatin accessibility in the RPE cells. We added the results and methods in the revised manuscript (page 12, paragraph 2 and page 25, paragraph 2).

Genotoxic response and Karyotyping

Reviewer response: The karyotype shown is at a very coarse level that would only reveal major genetic alterations. The authors should undertake a more detailed inspection such as through the use of SKY or through high density SNP genotyping to indicate that no smaller genetic changes have occurred.

RESPONSE: In the first round of review, the Reviewer criticized that the iPSC-derived RPE cells had multiple passages and “*The authors have not confirmed that karyotypic changes have not occurred at this passaging age*”. We already clarified that iPSC cells were differentiated to RPE cells in one passage. Therefore, the karyotypic changes are unlikely to occur during the process, and our karyotype result confirmed it. Second, even if any minor karyotypic change was induced during cell differentiation, we compared the RPE cells *before and after* the smoke treatment. Any genetic changes induced during the differentiation should thus not affect the interpretation of smoke treatment results, and the observed chromatin changes can only be attributed to smoke treatment. Third, we observed a global change in chromatin accessibility, which is unlikely due to local “smaller genetic changes”. Finally, no matter whether we observe the genetic changes or not, it will not alter our conclusion or strengthen our results.

=====

Reviewer #3 (Remarks to the Author):

The authors have addressed my concerns.

=====

Reviewer #4 (Remarks to the Author):

In this paper, Wang et al. performed ATAC-seq analysis between AMD patient samples and controls. Based on decreased ATAC-seq signals in AMD patient samples, Wang et al. concluded that the global chromatin accessibility is impaired in AMD patients. Mechanistically, the authors demonstrated that cigarette smoke extract treatment of RPE cells decreased ATAC-seq signal in iPS-cell-derived RPE cells. Overexpression of HDAC11 also lead to decreased ATAC-seq signal.

There are several strengths in this paper. First of all, human tissues in both disease and normal conditions were used to perform ATAC-seq. Those tissues are not easily accessible, and the data generated from these precious samples are informative to the general audience. Second, the global reduction of ATAC-seq signal is quite intriguing. Third, visualization of the data analysis was beautifully represented in many figures.

RESPONSE: We thank the Reviewer for the positive comments.

Despite the efforts to generate and analyze the data, the conclusions from the analysis need to be tuned down. The mechanisms suggested by the paper are not strongly supported by the data. Major concerns for the papers are listed as follows:

1) *decreased “global” ATAC-seq signal alone is not sufficient to demonstrate that the global chromatin accessibility is decreased. Since ATAC-seq relies on penetration of Tn5 transposase into the nuclei of the living cells, the permeability of the cell can directly influence the signal. In other words, the reduction of ATAC-seq signals in all three conditions (AMD, cigarette-extract treated RPE, and HDAC11 overexpression) could be explained alternatively by reduction of permeability. In addition, if global chromatin accessibility is indeed reduced in these conditions, one would predict that global gene expression would be reduced. However the authors clearly stated on page 14 that “no global decrease in RNA expression in late-stage AMD was observed”. Therefore this “global reduction in chromatin*

accessibility” could be a result of technical artifact, or it could have no direct causal relationship to gene expression.

RESPONSE: We thank the Reviewer for these insightful comments. However, it is unlikely that the observed ATAC-Seq signal is related to cell permeability. **First**, Tn5 transposase integration into the target genome was performed using nuclear extracts rather than intact cells (See Methods and ATAC-Seq protocol). Specifically, cells were lysed and nuclei were extracted before the transposase reaction was conducted. **Second**, we normalized the total number of reads for each sample to minimize potential other confounding effects. As a result, each sample has the same normalized transposase insertion events. Therefore, the global reduction of ATAC-Seq signals in AMD samples corresponds to the re-distribution of sequencing reads in the peak region to the non-peak regions. **Third**, the ATAC-Seq peak intensities were positively correlated with the gene expression level (Fig. 4e and Extended Data Fig. 5c), and genes associated with decreased ATAC-Seq signals tend to have low expression (Fig. 4f and Extended Data Fig. 5d), suggesting that the ATAC-Seq signals are highly associated with the downstream biological effects. **Finally**, ATAC-Seq has been widely used in many labs to profile chromatin accessibility. For example, a global increase in chromatin accessibility was observed during lung cancer metastatic progression³ using the same experimental approach employed in our studies. Taken these points together, we strongly suggest that the observed ATAC-Seq signals do indeed reflect chromatin accessibility in our work.

We realized that the sentence in the Discussion, “no global decrease in RNA expression in late-stage AMD was observed” is confusing and doesn’t add meaningful information. We deleted the sentence in the revised manuscript.

2) *For “Chronic” CSE treatment, The authors used 500ug/mL only for only 5 days at 2hours/day. Although CSE treatment has been reported in the literature for other cell types such as T cells and keratinocytes, RPE may have different response to this treatment. Did RPE cells have stress response? Did the proliferation rate of RPE cell change? All those factors could potentially affect ATAC-seq results, without directly impact chromatin accessibility during the interphase.*

RESPONSE: The Reviewer might have the concern that stress response or proliferation rate could affect

cell permeability, and thus alter the ATAC-Seq profile. However, as we discussed in the previous point, the cell permeability will not affect the ATAC-Seq profile. The observed ATAC-Seq signals should mainly reflect the chromatin accessibility. CSE is a complex chemical oxidant, and thus, it had to induce a stress response. That is the point of the experiment, to model a type of stress seen in AMD.

3) HDAC11 is generally considered as a class IV HDAC. It can shuttle between the cytoplasm and the nucleus. In RPE cells, is HDAC 11 predominantly localized to the nucleus? With HDAC11 overexpression, a logical experiment is to show that H3K27Ac level is altered. However this result was not included. To convincingly conclude that the “global” chromatin accessibility is decreased in AMD, additional evidence using alternative approaches should be included. For example, could staining of H3K27Ac be performed for tissue sections? Or could western be performed (if sufficient material available) to compare H3K27Ac levels?

RESPONSE: As suggested by the Reviewer, we performed HDAC11 staining in RPE cells and found that HDAC11 is predominantly localized in the nucleus. We included the results in Fig. 5e (page 13, paragraph 2).

We also performed western blot analysis for H3K27Ac in RPE cells in which HDAC11 was overexpressed. The result is included in Extended Data Fig. 7d (page 14, paragraph 1).

Based upon our above discussion of the first point, we hope that the Reviewer is now convinced that the observed ATAC-Seq signals are correlated with and reflect the chromatin accessibility. Regarding the additional experiments suggested by the reviewer, unfortunately, many of the eyes collected for this study were already used for other projects and we do not have sufficient materials to perform western blots to compare H3K27Ac levels between AMD and normal. We have not pursued immunostaining analysis of H3K27Ac because this technique would not give quantitative comparisons between the conditions.

Minor comments: Additional details should be included for the figures

1. Figure 1b, please consider adding the data of house keeping genes. Is the accessibility for house

keeping genes altered in AMD?

RESPONSE: Thanks for the insightful suggestion. We added housekeeping genes as examples (Extended Data Fig. 1c). We also performed systematic analysis of how housekeeping genes were associated with chromatin accessibility changes in AMD. Based on the list of housekeeping genes from a previous publication⁴, we examined the percentage of housekeeping genes that are associated with differentially accessible regions (DARs, significantly altered peaks in AMD). We found that housekeeping genes are significantly associated with DARs in the RPE but not with retinal DARs (Extended Data Fig. 5b). The housekeeping genes with essential cellular functions such as mitochondrion and cellular response to stress were significantly associated with DAR in the RPE. We added these results in the manuscript (page 10, paragraph 2).

2. Figure 1c, the scale bar needs a label such as “ATAC-seq signal intensity”.

RESPONSE: Thanks for the suggestion. We added the label.

2 Figure 2a upper panel: is this from three representative samples, or is this an aggregation of all data for each category?

RESPONSE: The upper panel shows an average signal of all data for each disease stage. The detailed data from each sample are shown in the bottom panel. We clarified this in the figure legend.

3 Figure 4 c, f, h: scale bars should be included to show the meanings of the colors.

RESPONSE: We added a scale bar to the figure.

References

1. Bhise NS, Wahlin KJ, Zack DJ, Green JJ. Evaluating the potential of poly(beta-amino ester) nanoparticles for reprogramming human fibroblasts to become induced pluripotent stem cells. *International journal of nanomedicine* **8**, 4641-4658 (2013).
2. Maruotti J, Wahlin K, Gorrell D, Bhutto I, Luttj G, Zack DJ. A simple and scalable process for the differentiation of retinal pigment epithelium from human pluripotent stem cells. *Stem cells translational medicine* **2**, 341-354 (2013).

3. Denny SK, *et al.* Nfib promotes metastasis through a widespread increase in chromatin accessibility. *Cell* **166**, 328-342 (2016).
4. Eisenberg E, Levanon EY. Human housekeeping genes, revisited. *Trends in genetics : TIG* **29**, 569-574 (2013).

Reviewers' comments:

Reviewer #2 (Remarks to the Author):

The authors are to be congratulated on attending to my suggestions. However, a few remaining points should be addressed.

Introduction

Page 1, para 1, line 1. Please change '...34 genetic risk loci involved in the regulation of the complement..' to ...34 genetic risk loci involved in multiple pathways including the regulation of the complement..'

Page 6, sentence beginning 'Moreover, most AMD samples are clearly separated from normal samples, suggesting..' to 'Moreover, most AMD samples are clearly separated from normal samples, especially for RPE, suggesting..'

Discussion.

Include a comment on your findings versus previous reports of IL17 and epigenetic marks.

Include a comment on why known AMD risk loci do not appear to be influenced by changes in chromatin.

Extended data table 2. Indicate which samples are early AMD and which are GA

Legend, line 1 'The name of sample is consisted of disease status...' does not make sense – please amend

Minor

A number of grammar issues, particularly in the methods should be attended to.

Reviewer #4 (Remarks to the Author):

Wang et al. provided very interesting data showing the alterations of ATAC-seq accessibilities in AMD. During the revision, the authors included a few additional data to support their major

conclusions: 1) global chromatin accessibility is decreased in age-related macular degeneration; 2) this decreased chromatin accessibility can be explained by elevated HDAC11.

The authors nicely demonstrated the decreased histone acetylation levels (down to 50%) with HDAC11 overexpression at 300%. It is important to keep in mind that HDAC11 only increases 0.24 by Log2 fold change (according to Extended Data Table 4), which is only about 18% increase at the mRNA level in RPE tissue. Therefore elevated HDAC11 may explain a small part of the overall decreased accessibility. It would be great if the authors can include a paragraph in the “Discussion” section to discuss other potential mechanisms. For example, perhaps HATs and HMTs may also have altered expression levels in AMD?

The authors strongly insisted that their ATAC-seq accessibility reflects chromatin accessibility based on 4 arguments. A. Nuclei extraction was performed for all cells. B. Normalization was done based on total number of read per sample. C. ATAC-seq peak intensities were positively correlated with gene expression level. D. ATAC-seq was used by Denny et al. for profile chromatin accessibility.

A. Nuclei extraction alone does not provide barrier-free access to chromatin for Tn5 transposase. First, the nucleus envelope serves as a strong barrier. Second, it is known that a lot of ATAC-seq reads are mapped to the mitochondria genome, which indicate that the nuclei extraction protocol by itself is unlikely to yield “pure” nuclei. It is not impossible that AMD may alter the properties of these barriers, impairing ATAC-seq efficiency.

B. Normalization is a tricky part of ATAC-seq data analysis. Indeed the routine is to normalize by the total sequencing depth; however this normalization method does not take the cell numbers into consideration. For RNA-seq, many labs are now using spike in controls. It is understandable that spike-in may be difficult for this paper.

C. The correlation between ATAC-seq peak intensity and gene expression level. The authors demonstrated the positive correlation between ATAC-seq intensity and gene expression level. Intriguingly, the authors also removed their statement “no global decrease in RNA expression in late-stage AMD was observed” during the revision. Does this imply that the global RNA expression should be decreased in AMD?

D. Comparison with the data present by Denny et al.. One the key differences in terms of ATAC-seq data presentation is that Denny et al. showed many examples of large genomic regions, ranging from 10kb to several Mb in length. The ATAC-seq signals in both intergenic regions and genes were shown. In particular in Figure 2B from Denny et al, the authors demonstrated many regions with

comparable accessibility between two conditions, as well as a subset of regions gained accessibility in one condition. It is well known that the majority of ATAC-seq signals come from intergenic regulatory regions. However in this paper from Wang et al., the author only presented very narrow regions of ATAC-seq signals near TSS of a few representative genes. There is no label for the Y-axis. Therefore it is very hard to judge the quality of their ATAC-seq data. Can the authors simply provide 3 examples of 1-Mb genomic regions to better 1) demonstrate the quality of their ATAC-seq data and 2) the selectivity of altered ATAC-seq accessibility along the genome as demonstrated by Denny et al.?

Reviewer #2 (Remarks to the Author):

The authors are to be congratulated on attending to my suggestions. However, a few remaining points should be addressed.

Introduction

Page 1, para 1, line 1. Please change ‘...34 genetic risk loci involved in the regulation of the complement..’ to ...34 genetic risk loci involved in multiple pathways including the regulation of the complement...’

RESPONSE: We thank the reviewer for this suggestion. We have made the change as requested.

Page 6, sentence beginning ‘Moreover, most AMD samples are clearly separated from normal samples, suggesting..’ to ‘Moreover, most AMD samples are clearly separated from normal samples, especially for RPE, suggesting...’

RESPONSE: We thank the reviewer for this suggestion. We have made the change as requested.

Discussion.

Include a comment on your findings versus previous reports of IL17 and epigenetic marks.

Include a comment on why known AMD risk loci do not appear to be influenced by changes in chromatin.

RESPONSE: We added more discussion on these two points.

Point 1: “It is known that environmental factors contribute to the development of AMD 20-22. These environmental factors may alter epigenetic marks, which in turn can lead to broad biological consequence 23. DNA methylation in blood or retina has been studied in AMD 9-12, with one example of an AMD-associated change being the hypomethylation of the *IL17RC* promoter that is observed in peripheral blood leukocytes 10. However, the finding remains controversial 11. Overall, the changes of DNA methylation in AMD are quite subtle.”

Point 2: “AMD risk loci are not significantly over-represented in the identified DARs, suggesting that the observed differences in chromatin accessibility are unlikely to result from local AMD-associated genetic variants. However, our study does not exclude the possibility that the chromatin accessibility is associated with other genetic variants. To fully investigate the possible interplay between the two factors, simultaneous measurement of chromatin accessibility and genetic variants will need to be conducted in a much larger number of samples.”

Extended data table 2. Indicate which samples are early AMD and which are GA

RESPONSE: We added the information to the table.

Legend, line 1 ‘The name of sample is consisted of disease status...’ does not make sense – please amend

RESPONSE: We made the changes.

A number of grammar issues, particularly in the methods should be attended to.

RESPONSE: We carefully re-read the manuscript and corrected the few grammatical errors we found.

Reviewer #4 (Remarks to the Author):

Wang et al. provided very interesting data showing the alterations of ATAC-seq accessibilities in AMD. During the revision, the authors included a few additional data to support their major conclusions: 1) global chromatin accessibility is decreased in age-related macular degeneration; 2) this decreased chromatin accessibility can be explained by elevated HDAC11.

The authors nicely demonstrated the decreased histone acetylation levels (down to 50%) with HDAC11 overexpression at 300%. It is important to keep in mind that HDAC11 only increases 0.24 by Log2 fold change (according to Extended Data Table 4), which is only about 18% increase at the mRNA level in RPE tissue. Therefore elevated HDAC11 may explain a small part of the overall decreased accessibility. It would be great if the authors can include a paragraph in the "Discussion" section to discuss other potential mechanisms. For example, perhaps HATs and HMTs may also have altered expression levels in AMD?

RESPONSE: We thank the reviewer for this suggestion. We added more discussion on this point. Specifically, "Our study demonstrated that upregulated HDAC11 expression might be partially responsible for the observed changes in chromatin accessibility in AMD. However, the effect of HDAC11 on chromatin accessibility is limited, suggesting that other factors (e.g. HATs) may also contribute to the observed DARs. Beyond changes in expression of these general chromatin and DNA modification enzymes, which do not possess sequence specificity, we hypothesize that altered expression of specific transcription factors (TFs) that play a role in guiding these enzymes to specific genomic loci may also account for the observed changes in chromatin accessibility in AMD."

The authors strongly insisted that their ATAC-seq accessibility reflects chromatin accessibility based on 4 arguments. A. Nuclei extraction was performed for all cells. B. Normalization was done based on total number of read per sample. C. ATAC-seq peak intensities were positively correlated with gene expression level. D. ATAC-seq was used by Denny et al. for profile chromatin accessibility.

A. Nuclei extraction alone does not provide barrier-free access to chromatin for Tn5 transposase. First, the nucleus envelope serves as a strong barrier. Second, it is known that a lot of ATAC-seq reads are mapped to the mitochondria genome, which indicate that the nuclei extraction protocol by itself is unlikely to yield "pure" nuclei. It is not impossible that AMD may alter the properties of these barriers, impairing ATAC-seq efficiency.

B. Normalization is a tricky part of ATAC-seq data analysis. Indeed the routine is to normalize by the total sequencing depth; however this normalization method does not take the cell numbers into consideration. For RNA-seq, many labs are now using spike in controls. It is understandable that spike-in may be difficult for this paper.

C. The correlation between ATAC-seq peak intensity and gene expression level. The authors demonstrated the positive correlation between ATAC-seq intensity and gene expression level. Intriguingly, the authors also removed their statement “no global decrease in RNA expression in late-stage AMD was observed” during the revision. Does this imply that the global RNA expression should be decreased in AMD?

D. Comparison with the data present by Denny et al.. One the key differences in terms of ATAC-seq data presentation is that Denny et al. showed many examples of large genomic regions, ranging from 10kb to several Mb in length. The ATAC-seq signals in both intergenic regions and genes were shown. In particular in Figure 2B from Denny et al, the authors demonstrated many regions with comparable accessibility between two conditions, as well as a subset of regions gained accessibility in one condition. It is well known that the majority of ATAC-seq signals come from intergenic regulatory regions. However in this paper from Wang et al., the author only presented very narrow regions of ATAC-seq signals near TSS of a few representative genes. There is no label for the Y-axis. Therefore it is very hard to judge the quality of their ATAC-seq data. Can the authors simply provide 3 examples of 1-Mb genomic regions to better 1) demonstrate the quality of their ATAC-seq data and 2) the selectivity of altered ATAC-seq accessibility along the genome as demonstrated by Denny et al.?

RESPONSE: The reviewer still has the concern whether the ATAC-Seq observed in this study reflects changes in chromatin accessibility observed in AMD. The ATAC-Seq is a widely received technique in the field. Since it was published in 2013 (Buenrostro et al Nature Methods, 2013), many labs have applied it to study chromatin accessibility in many systems. Indeed, more than 130 papers have been published that use this technique. ATAC-Seq results have been directly compared to the previous gold standard approach, DNase I hypersensitivity sites (DNase HS), which was based on identification of DNase I cleavage sites in open chromatin regions. The comparison shows that ATAC-Seq highly correlates with DHSs (see Figure below from Buenrostro et al paper). Furthermore, comparison with results obtained using FAIRE-seq, which doesn't require permeabilization of cells or isolation of nuclei, also showed that ATAC-Seq results closely match FAIRE-seq signals. With that said, we now added a brief discussion of potential limitations of this technology, specifically: “ATAC-Seq is a widely used approach to detect chromatin accessibility. However, one potential confounding variable is nuclear envelope permeability, which could possibly influence the ATAC-Seq signals. Since this analysis was performed using nuclear extracts, it is conceivable that AMD might induce differences in nuclear permeability. While multiple evidence presented in this study strongly support our interpretation that ATAC-Seq signals reflect chromatin accessibility, it remains possible that AMD might affect nucleic envelope permeability.”

The reviewer also suggested using spike-in controls in ATAC-Seq. While it is feasible to add (spike in) controlled amounts of molecules in RNA-seq libraries, it is technically challenging to have a controlled number of transposase that are integrated in the genome. Reviewer also agreed that “*It is understandable that spike-in may be difficult for this paper*”.

We removed the statement “no global decrease in RNA expression in late-stage AMD was observed”, which was accidentally included by one colleague during the many rounds of manuscript revision. We did not pay much attention on this part and thank the reviewer for pointing this out.

Thanks to the reviewer’s great suggestion, we also generated a few examples of ATAC-Seq signals in large chromosomal regions. As shown in the Figure below, the top panel shows a 100 Mb genomic region. In this region, we can find many genomic domains (1-2Mb size), in which the ATAC-Seq signals in AMD samples are significantly lower than those in normal samples. The bottom panel highlights a few of these domains (2Mb each). The first three regions show the global reduction of ATAC-Seq signals in AMD, while the fourth example shows both reduced peaks in AMD and non-affected peaks. We included the new result as Extended data Fig. 4. As requested by the reviewer, we also added the Y-axis in Figure 2A.

Reviewer #4 (Remarks to the Author):

The authors did a nice job in adding open discussions on the potential technical limitation/bias of ATAC-seq as well as representative data tracks.

Many potential readers may only scan through the title and abstract. It would be great if the authors can specify the ATAC-seq method in both title and abstract.

Reviewer #4 (Remarks to the Author):

The authors did a nice job in adding open discussions on the potential technical limitation/bias of ATAC-seq as well as representative data tracks.

Many potential readers may only scan through the title and abstract. It would be great if the authors can specify the ATAC-seq method in both title and abstract.

RESPONSE: We thank the reviewer for the suggestion. We have made the change as requested.